# The autophagy receptor NBR1 directs the clearance of photodamaged chloroplasts

Han Nim Lee[1,2], Jenu Varghese Chacko[1], Ariadna Gonzalez Solís[1,2], Kuo-En Chen[3], Jessica AS Barros[3], Santiago Signorelli[4,5], A Harvey Millar[4], Richard David Vierstra[3], Kevin W Eliceiri[1,6,7,8], Marisa S Otegui[1,2]*

[1]Center for Quantitative Cell Imaging, University of Wisconsin-Madison, Madison, United States; [2]Department of Botany, University of Wisconsin-Madison, Madison, United States; [3]Department of Biology, Washington University in St. Louis, Saint Louis, United States; [4]ARC Centre of Excellence in Plant Energy Biology, School of Molecular Sciences, The University of Western Australia, Perth, Australia; [5]Department of Plant Biology,School of Agronomy, Universidad de la República, Montevideo, Uruguay; [6]Department of Medical Physics, University of Wisconsin-Madison, Madison, United States; [7]Department of Biomedical Engineering, University of Wisconsin-Madison, Madison, United States; [8]Morgridge Institute for Research, Madison, United States

*For correspondence:
otegui@wisc.edu

Competing interest: The authors declare that no competing interests exist.

**Abstract** The ubiquitin-binding NBR1 autophagy receptor plays a prominent role in recognizing ubiquitylated protein aggregates for vacuolar degradation by macroautophagy. Here, we show that upon exposing *Arabidopsis* plants to intense light, NBR1 associates with photodamaged chloroplasts independently of ATG7, a core component of the canonical autophagy machinery. NBR1 coats both the surface and interior of chloroplasts, which is then followed by direct engulfment of the organelles into the central vacuole via a microautophagy-type process. The relocalization of NBR1 into chloroplasts does not require the chloroplast translocon complexes embedded in the envelope but is instead greatly enhanced by removing the self-oligomerization mPB1 domain of NBR1. The delivery of NBR1-decorated chloroplasts into vacuoles depends on the ubiquitin-binding UBA2 domain of NBR1 but is independent of the ubiquitin E3 ligases SP1 and PUB4, known to direct the ubiquitylation of chloroplast surface proteins. Compared to wild-type plants, *nbr1* mutants have altered levels of a subset of chloroplast proteins and display abnormal chloroplast density and sizes upon high light exposure. We postulate that, as photodamaged chloroplasts lose envelope integrity, cytosolic ligases reach the chloroplast interior to ubiquitylate thylakoid and stroma proteins which are then recognized by NBR1 for autophagic clearance. This study uncovers a new function of NBR1 in the degradation of damaged chloroplasts by microautophagy.

## Editor's evaluation

This fundamental work substantially advances our understanding of the role of the ubiquitin-binding NBR1 autophagy receptor protein. The work proposed a new function of this protein in the degradation of damaged chloroplasts upon exposing Arabidopsis plants to intense light. The evidence supporting the conclusions is compelling, using a combination of genetics, proteomics and state-of-the-art microscopy. The work will be of broad interest to cell biologists and biochemists.

## Introduction

Autophagy is a process by which cytoplasmic contents including organelles, individual proteins, protein complexes and cytosolic aggregates, collectively called autophagic cargo, are delivered to vacuoles (plants and yeast) and lysosomes (animals) for degradation (*Mizushima et al., 1998*). In plants, autophagy most commonly occurs through the formation of cargo-sequestering double-membrane-bound organelles called autophagosomes (macroautophagy) or through the direct engulfment of cargo by the vacuolar membrane (microautophagy). Although the molecular underpinnings of microautophagy are poorly understood, the machinery behind macroautophagy involves more than 40 ATG (Autophagy Related) proteins whose actions are regulated by upstream phosphorylation events ultimately leading to formation of autophagosomes decorated with a conjugate of ATG8 bearing phosphatidylethanolamine (PE) (*Xie et al., 2008*). This lipidation is mediated by an enzymatic cascade sequentially involving the activating enzyme ATG7, the conjugating enzyme ATG3, and a ligase complex comprising an ATG5-ATG12 conjugate complexed with ATG16 (*Ohsumi, 2001*). The resulting ATG8-PE adduct is not only required for autophagosomes assembly but also, through its interaction with a host of autophagic receptors, for the selection of appropriate autophagic cargo (*Noda et al., 2008*).

There are several selective autophagy receptors that specifically recognize ubiquitylated cargo. Among them, metazoan SQSTM1/p62 (Sequestosome 1) and NBR1 (NEIGHBOR OF BRCA1 gene 1) promote the accretion of ubiquitylated proteins into larger condensates which are then encapsulated by autophagosomes for macroautophagic clearance (aggrephagy; *Bjørkøy et al., 2005*; *Komatsu et al., 2007*; *Nezis et al., 2008*; *Turco et al., 2021*; *Rasmussen et al., 2022*). The PB1 (Phox and Bem1 domain) domain present in both SQSTM1 and NBR1 mediate their mutual interaction and oligomerization into helical filaments (*Ciuffa et al., 2015*) which then promote the aggregation of ubiquitylated species (*Jakobi et al., 2020*; *Turco et al., 2021*). In addition, mammalian SQSTM1 and NBR1 share a zinc-finger domain (ZZ) that can bind N-terminally arginylated proteins, polyubiquitylated proteins, and other cargo (*Cha-Molstad et al., 2015*; *Kwon et al., 2018*; *Wang et al., 2021*), a ubiquitin-associated (UBA) domain with affinity for ubiquitin, and an ATG8-interacting motif (AIM) sequence that binds ATG8 (*Seibenhener et al., 2004*; *Ichimura et al., 2008*; *Zientara-Rytter et al., 2011*; *Sun et al., 2022a*). NBR1, but not SQSTM1, also contains a Four-Tryptophan (FW) domain, which at least in some fungal species helps recognize cargo for selective autophagy. The plant NBR1 proteins uniquely harbor two UBA domains but only the C-terminal sequence (UBA2) binds ubiquitin (*Svenning et al., 2011*). Through these collective features, SQSTM1 and NBR1 can mediate selective autophagy of cargo in both ubiquitin and ubiquitin-independent manners. Most non-metazoan species encode only NBR1, whereas metazoans can express either or both SQTM1 and NBR1 (*Svenning et al., 2011*).

In plants, NBR1 has been connected genetically to numerous physiological processes (*Zhang and Chen, 2020*). For example, it modulates tolerance to heat stress through at least two mechanisms; recognition and sorting for degradation of proteotoxic ubiquitylated aggregates that accumulate at high temperatures (*Zhou et al., 2013*; *Zhou et al., 2014*), and the negative regulation of heat stress memory by mediating the clearance of heat-shock-related chaperones and their co-factors (*Thirumalaikumar et al., 2021*). *Arabidopsis* NBR1 also targets for autophagic clearance: (i) the exocyst subunit EXO70E2 and its associated organelle EXPO (*Ji et al., 2020*), (ii) misfolded protein aggregates (*Jung et al., 2020*), and (iii) viral capsid proteins (*Hafrén et al., 2017*) and pathogenic bacterial effectors (*Dagdas et al., 2016*; *Dagdas et al., 2018*; *Üstün et al., 2018*; *Leong et al., 2022*). Remarkably, *Arabidopsis* null *nbr1* mutants develop normally under favorable growth conditions and are still able to execute general autophagy (*Jung et al., 2020*) and the selective clearance of certain organelles such as peroxisomes (*Young et al., 2019*). However, the mutants are hypersensitive to heat, drought, oxidative, and salt stress and over-accumulate cytoplasmic protein aggregates (*Zhou et al., 2013*). Taken together, NBR1 appears to be required for some but not all autophagy-dependent events, consistent with a role in selective autophagy.

Chloroplast turnover involves multiple routes that are dependent on autophagy and/or the ubiquitin-proteasome system. Several ATG8-dependent autophagic routes control the piecemeal turnover of chloroplast components *via* Rubisco-containing bodies (*Chiba et al., 2003*; *Ishida et al., 2008*; *Spitzer et al., 2015*), ATI1-PS (ATG8-INTERACTING PROTEIN 1) bodies (*Michaeli et al., 2014*), and SSLG (small starch-like granule) bodies (*Wang et al., 2013*) as well as the engulfment of whole

photodamaged chloroplasts through microautophagy (*Izumi et al., 2017*). Outer envelope proteins, including components of the outer envelope translocon complex (TOC), can be ubiquitylated by chloroplast membrane-localized ubiquitin E3 ligase SP1 (SUPPRESOR OF PPI1 LOCUS 1) and extracted from the envelope membrane by the β-barrel channel protein SP2 and the AAA +ATPase CDC48 for degradation by the 26 S proteasome in a process named chloroplast-associated degradation (CHLORAD) (*Ling et al., 2012*; *Ling et al., 2019*). The cytosolic E3 ligase PLANT U-BOX4 (PUB4) also ubiquitylates chloroplast envelope proteins in response to oxidative stress (*Woodson et al., 2015*). More recently, proteins within the chloroplast lumen (e.g. thylakoid and stroma proteins) were also shown to be targeted by ubiquitylation for break down *via* the proteasomes under oxidative stress (*Li et al., 2022*; *Sun et al., 2022b*). However, it remains unclear how ubiquitylation occurs inside chloroplasts.

Here, we show that NBR1 associates with photodamaged chloroplasts *via* its ubiquitin-binding UBA domain and mediates their vacuolar degradation by an autophagic pathway independent of ATG7, and therefore, of ATG8 lipidation. NBR1 associates with the surface and interior of chloroplasts without the need for intact translocon complexes within the outer and inner membranes. We proposed that photodamaged chloroplasts lose structural integrity of their envelopes, thus allowing access of cytosolic components such as the ubiquitylation machinery and NBR1 into the plastid interior for subsequent microautophagic clearance.

## Results

### NBR1 associate with chloroplasts upon exposure to high light

To determine whether the autophagy receptor NBR1 is involved in chloroplast turnover upon photoradiation damage, we imaged by confocal microscopy the NBR1-GFP fusion protein expressed under the control of the NBR1 promoter [*ProNBR1:NBR1-GFP*; (*Hafrén et al., 2017*; *Thirumalaikumar et al., 2021*)] in seedlings grown under low light (LL; 40 µmol m$^{-2}$ s$^{-1}$) at 22 °C and then exposed to high light (HL; 1500 µmol m$^{-2}$ s$^{-1}$) at 12 °C for 2 h, with a focus on the sub-adaxial epidermal mesophyll layer (mesophyll cells under the cotyledon adaxial epidermis) exposed to HL. Under LL, NBR1 was typically found in cytosolic puncta within cotyledons that likely represent aggresome condensates (*Svenning et al., 2011*; *Jung et al., 2020*) and did not colocalize with chloroplasts seen by chlorophyll autofluorescence (*Figure 1*). After exposing seedlings to HL and allowing them to recover under LL for 24 hr, 2% of the chloroplasts in HL-exposed mesophyll cells became heavily decorated with NBR1-GFP (*Figure 1A and B*). NBR1-GFP either coated the surface of these chloroplasts or, in a few cases, localized inside (*Figure 1A*). Some NBR1-GFP signal in hypocotyl cells was also associated with stromules (*Figure 1A*).

To determine whether NBR1-GFP associated with photodamaged chloroplasts, we measured chlorophyll intrinsic fluorescence from seedlings either kept under LL or left to recover after HL exposure. In cotyledons exposed to HL, chloroplasts not labeled by NBR1-GFP had chlorophyll intensity values similar to those of control chloroplasts kept under LL. In contrast, NBR1-GFP-decorated chloroplasts showed a significant decrease in chlorophyll fluorescence intensity (*Figure 1C*), consistent with chlorophyll breakdown after photodamage (*Nakamura et al., 2018*). As an indicator of chloroplast photodamage, we quantified the chlorophyll fluorescence intensity ratio measured at 517 and 683 nm (*Nakamura et al., 2018*), and found a statistically significant increase in this ratio for NBR1-GFP-decorated chloroplasts after HL exposure (*Figure 1D*).

Previous studies showed that the recruitment of ATG8 to chloroplasts after HL exposure depends on the canonical ATG machinery (*Nakamura et al., 2018*). Consequently, we tested whether this was also the case for NBR1 by analyzing seedlings expressing mCherry-NBR1 under the control of the *UB10* promoter in the *nbr1-2* (*Zhou et al., 2013*; *Jung et al., 2020*) and *atg7-2* (*Chung et al., 2010*) mutant backgrounds. Upon HL exposure, we detected by confocal microscopy mCherry-NBR1 associated with chloroplasts in both *nbr1* and *atg7* cotyledon mesophyll cells (*Figure 1E and F*). In both genotypes, the mCherry-NBR1 signal coated the chloroplast surface (open arrows in *Figure 1E*) as well as its interior (solid arrows in *Figure 1E*), indicating that ATG7, and by inference ATG8 lipidation, were not required for recruiting NBR1 to chloroplasts upon HL exposure.

To confirm that NBR1 was indeed internalized into chloroplasts, we examined the ultrastructural features of chloroplasts under HL conditions by transmission electron microscopy and localized NBR1

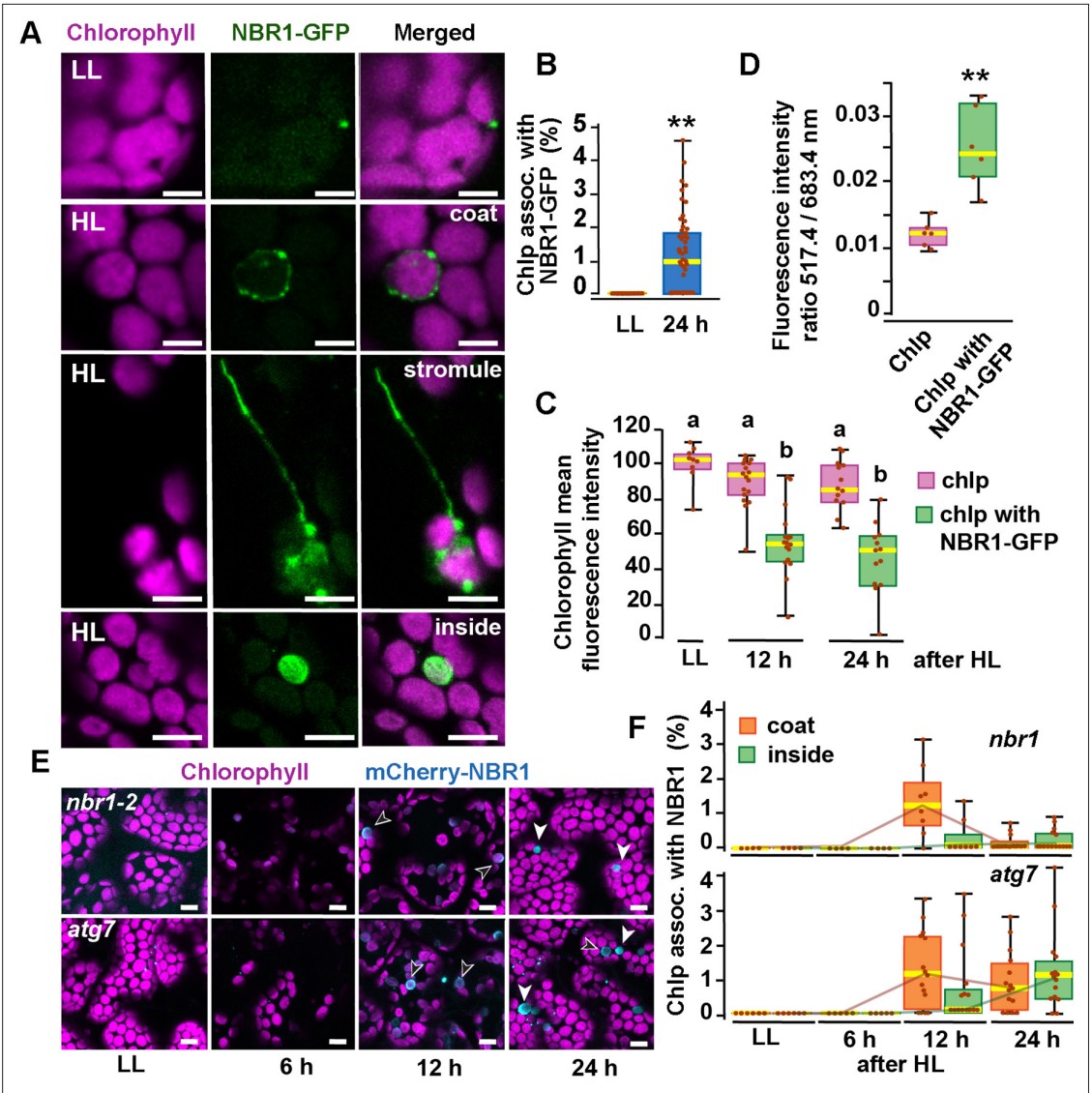

**Figure 1.** NBR1 associates with chloroplasts after HL exposure. (**A**) Confocal imaging of NBR1-GFP and chlorophyll autofluorescence in cotyledons and hypocotyl cells of 8-day-old wild-type seedlings grown under low light (LL, 40 μmol m⁻² s⁻¹) or left to recover for 24 hr after exposure to 2 hr HL conditions (HL, 1,500 μmol m⁻² s⁻¹) at 12 °C. After HL exposure, NBR1 either coated the surface of chloroplasts and stromules or localized inside chloroplasts. (**B**) Box and whisker plots represent the percentage of chloroplast associated with NBR1-GFP in 8-day-old seedlings grown under LL or 24 hr after HL exposure. At least 35 confocal images from 7 to 12 cotyledons were analyzed for each condition. (**C**) Box and whisker plots showing chlorophyll mean intensity from chloroplast with and without NBR1-GFP in cotyledons from 8-day-old seedling grown under LL or exposed to HL and left to recover for 12 hr or 24 hr. Representative experiment showing data from at least 5 randomly selected chloroplasts for each condition. (**D**) Ratio of chlorophyll fluorescence intensities at 517.4 m and 683.4 nm. Representative experiment showing data from 6 chloroplasts with or without NBR1-GFP from 8 day old cotyledons 24 hr after HL exposure. Different letters on the graph indicate significant difference (*P*<0.05) calculated by one-way ANOVA followed by Tukey's test. (**E**) Confocal imaging of cotyledons from 8-day-old seedling expressing mCherry-NBR1 in *nbr1* and *atg7* plants grown under LL or exposed to HL and left to recover for 6, 12, and 24 hr. Hollow arrowheads and filled arrowheads indicate the mCherry-NBR1 coats and inside chloroplasts, respectively. (**F**) Box and whisker plots showing the percentage of chloroplasts associated with mCherry-NBR1 as coats (orange) or inside chloroplasts (green) under LL, and at the indicated recovery times after HL exposure. The top and bottom plots show measurements from *nbr1* and *atg7*, respectively. Representative experiment analyzing between 4 and 15 fields from 3 to 6 cotyledons for each condition and genotype. Box and whisker plots in B, C, D, and F display the variation in data through quartiles; the middle line indicates the median and whiskers show the upper and lower fences. Asterisks in B and D denote significant differences based on t-tests (**, p<0.01). Scale bars = 5 μm in A and E.

with anti-NBR1 antibodies. First, we analyzed the structural alterations of chloroplasts after 24 hr exposure to HL in wild-type, *atg7*, and *nbr1* cotyledons processed by high-pressure frozen/freeze substitution. Based on the degree of structural integrity, we found three morphologically distinct chloroplast types often in the same cell although with different frequencies. Type-1 chloroplasts had normal thylakoids and electron-dense stroma; Type-2 chloroplasts had dilated and lighter stroma with thylakoid membranes partially disorganized and often displaced to one side of the chloroplast; and Type-3 chloroplasts contained highly disorganized thylakoids, light stroma, and clear signs of chloroplast envelope rupture (*Figure 2A–C*).

Type-3 chloroplasts were significantly more abundant in the *atg7* cotyledons, whereas their frequency in *nbr1* cotyledons was indistinguishable from that in wild-type cotyledons (*Figure 2D*). Using anti-NBR1 antibodies (*Figure 2—figure supplement 1*), we performed immunogold labeling to detect the native NBR1 protein in the three types of chloroplasts from wild-type and *atg7* cotyledons exposed to HL, in this case using *nbr1* seedlings grown under similar conditions as a negative control (*Figure 2E and F*). Whereas we did not detect labeling of NBR1 in the cytosol, all chloroplasts in wild-type and *atg7* seedlings exposed to HL showed significantly higher labeling than those seen in the *nbr1* cotyledons (*Figure 2E and F*).

Corroborating the NBR1-GFP and mCherry-NBR1 confocal imaging results, endogenous NBR1 was detected on the surface and inside wild-type and *atg7* chloroplasts (*Figure 2E*). Among the three types of chloroplasts, Type-3 chloroplasts, which were most abundant in *atg7* cotyledons (*Figure 2D*), showed the heaviest internal labeling, both on thylakoids and on the stroma (*Figure 2F*). As Type-3 chloroplasts showed disorganized thylakoids, this labeling is consistent with the preferential recruitment of NBR1-GFP to damaged chloroplasts as judged by their decreased levels of chlorophyll autofluorescence (*Figure 1C*).

To further validate the association of NBR1 with photodamaged chloroplasts, we isolated chloroplasts from 4-week-old wild-type and *atg7* mutant plants kept under LL or exposed to HL conditions and allowed to recover for 24 hr (*Figure 2G*). We assessed the purity of our chloroplast fraction by testing the enrichment of chloroplast proteins such as TIC40 (inner envelope) and anti-LHCIIb (thylakoid), and the depletion of the cytosolic fructose bisphosphatase (FBPase). NBR1 was barely detected in either the total extract or the chloroplast fraction from wild-type plants kept under LL (*Figure 2G*). However, after HL exposure, NBR1 became much more abundant in the chloroplast fraction. The association of NBR1 with chloroplasts under both LL and HL was also apparent in *atg7* seedlings (*Figure 2G*), further confirming that ATG7 is not required for recruiting NBR1 to photodamaged chloroplasts.

## ATG8 and NBR1 are recruited to different populations of damaged chloroplasts

ATG8 was previously reported to coat photodamaged chloroplasts in *Arabidopsis* (*Nakamura et al., 2018*). Since NBR1 interacts with ATG8, we tested whether NBR1 and ATG8 were recruited to the same chloroplast population. We used seedlings expressing both mCherry-NBR1 and GFP-ATG8, exposed them to HL, and then imaged them during a 24 hr recovery window (*Figure 3A and B*). As expected, neither mCherry-NBR1 or GFP-ATG8 associated with chloroplasts under LL conditions. However, after the HL treatment, the chloroplast association of both proteins became evident at 6 hr during recovery. By 12 hr after HL exposure, approximately 4% and 5% of the total mesophyll chloroplasts were decorated by either mCherry-NBR1 or GFP-ATG8, respectively, but remarkably only 1% of chloroplasts were decorated with both (*Figure 3A and B*). A similar trend was observed 24 hr after HL exposure; approximately 7% of the chloroplasts were labeled with GFP-ATG8, 5.5% were labeled with mCherry-NBR1 but only 2% of the chloroplasts were associated with both (*Figure 3A and B*). This dichotomy suggests that NBR1 and ATG8 associate with unique populations of chloroplasts, consistent with their distinct dependence on the ATG machinery for chloroplast recruitment.

To further assess a functional disconnection between ATG8 and NBR1 in the degradation of photodamaged chloroplasts, we imaged GFP-ATG8A in *nbr1*, *atg7*, and wild-type seedlings exposed to HL (*Figure 3C and D*). As previously reported (*Nakamura et al., 2018*), GFP-ATG8 failed to label photodamaged chloroplasts in *atg7* cotyledons. Compared to wild type, we detected a significant decrease in the proportion of chloroplasts decorated by GFP-ATG8A in the *nbr1* mutant at 6 hr during recovery from HL exposure; however, by 24 hr, similar proportions of both wild type and *nbr1*

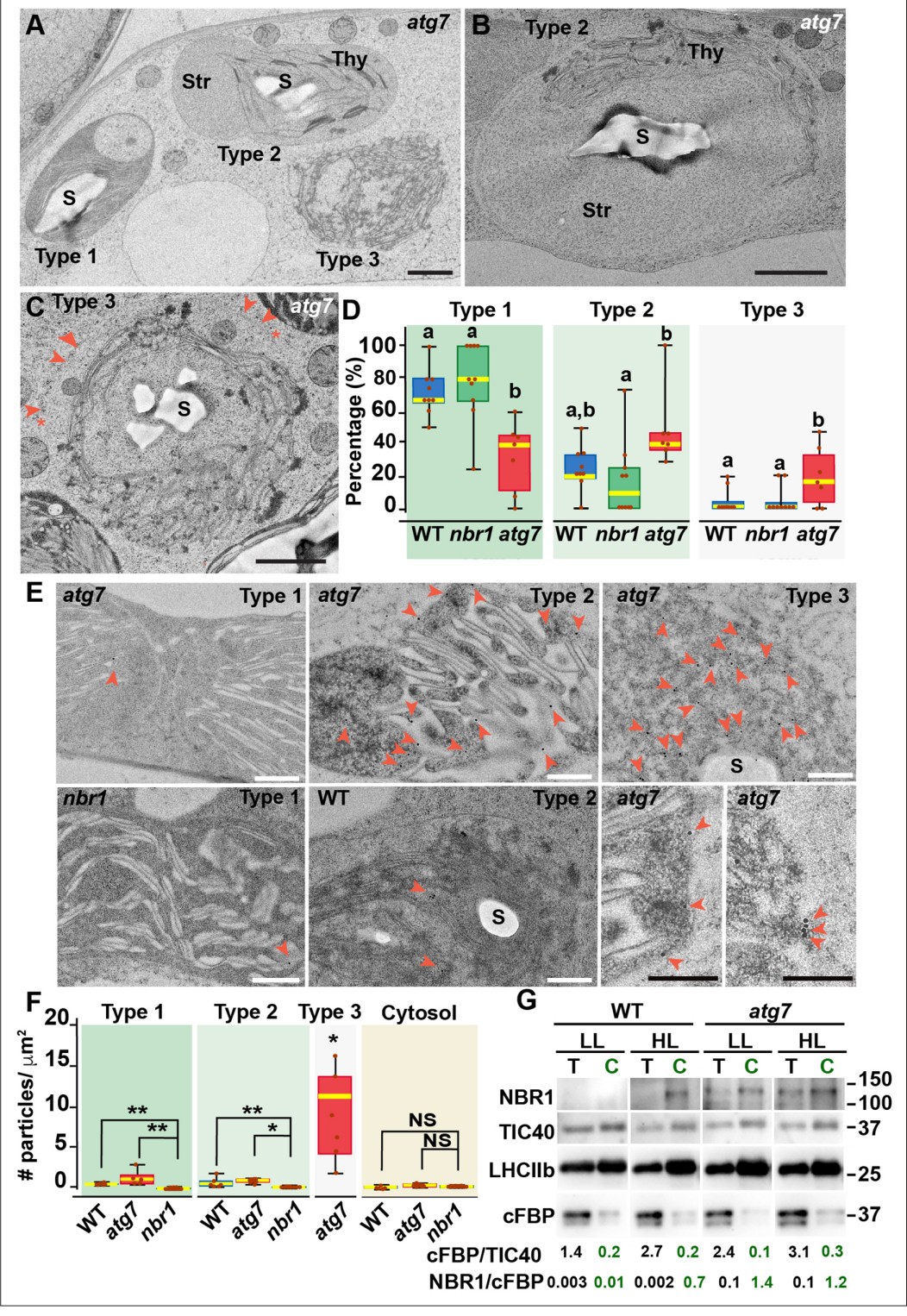

**Figure 2.** Ultrastructure of chloroplasts in wild-type, *atg7*, and *nbr1* cotyledons 24 hr after HL exposure. (**A**) Transmission electron micrograph of a high-pressure frozen/freeze-substituted *atg7* cotyledon mesophyll cell from 8-day-old seedlings exposed to HL and left to recover for 24 hr. Three different types of chloroplasts based on their structural integrity are seen. Type-1 chloroplasts with electron dense stroma and tightly appressed thylakoids, Type-2 chloroplasts with lighter stroma and partially disorganized thylakoids, and Type-3 chloroplasts with ruptured envelopes, disorganized thylakoid membranes and a stroma region with similar electron density and appearance to the cytoplasm. (**B, C**) Representative Type-2 (**B**) and Type-3 (**C**) chloroplasts in an *atg7* mesophyll

*Figure 2 continued on next page*

*Figure 2 continued*

cells. Note in (**C**) that the outer and inner envelopes (arrowheads) are disrupted in several sites (asterisks) exposing the interior of the chloroplast, including thylakoid membranes to the cytosol. (**D**) Box and whisker plots showing the percentage of Type-1,–2, and –3 chloroplasts per mesophyll cell section in wild-type Col-0 (WT), *nbr1*, and *atg7* cotyledons. Different letters on the graph indicate significant difference (p<0.05) calculated by one-way ANOVA followed by Tukey's test. Between 7 and 10 cells from two cotyledons of each genotype were used for this analysis. (**E**) Immunogold labeling with anti NBR1 antibodies on chloroplasts of WT, *nbr1*, and *atg7* mutant mesophyll cells exposed to HL followed by 24 hr recovery. Red arrowheads indicate gold particles on chloroplasts. (**F**) Quantification of anti-NBR1 gold labeling on Type-1, –2, and –3 chloroplasts and cytoplasm from WT, *atg7*, and *nbr1* mutant mesophyll cells exposed to HL. A t-test was used to compare values between mutant and WT samples; * and ** indicate p<0.05 and p<0.01, respectively. Between 5 and 11 chloroplasts or cytoplasmic regions from 2 cotyledons of each genotype were used for quantification. (**G**) Immunoblot detection of NBR1, TIC40 (chloroplast inner envelope protein), LHCIIb (thylakoid protein), and cFBP (cytosolic protein) in total protein extracts (T) and chloroplast protein fraction (**C**) from 4-week-old WT and *atg7* plants grown under LL or exposed to HL and let recover for 24 hr. The numbers below indicate the ratios cFBP/TIC40 and NBR1/cFBPase based on the quantification of the western blots. The figure shows a representative set of western blots. The experiment was repeated twice. Box and whisker plots in D and F show the variation in data through quartiles; the middle line indicates the median and whiskers show the upper and lower fences. S, starch; St, stroma; Thy, thylakoids. Scale bars: 1 μm in A, B, C; 500 nm in E.

The online version of this article includes the following source data and figure supplement(s) for figure 2:

**Source data 1.** G.Original files of full raw unedited blots and figure with uncropped blots.

**Figure supplement 1.** Uncropped immunoblot of total proteins from wild type plants expressing NBR1-GFP, and *atg7-1* and *nbr1* mutants using anti-NBR1 antibodies (*Jung et al., 2020*).

**Figure supplement 1—source data 1.** Original files of full raw unedited blots.

---

chloroplasts were coated by GFP-ATG8A (*Figure 3C and D*). Taken together, these results showed that NBR1 and ATG8A are recruited to different populations of photodamaged chloroplasts and that NBR1 is only partially required for the early association of GFP-ATG8A with these organelles.

## NBR1-decorated chloroplasts are delivered to vacuoles in an ATG7-independent manner

Previous studies have shown that ATG8-associated chloroplasts are delivered to vacuoles through a microautophagic process that relies on the canonical ATG machinery (*Izumi et al., 2017*; *Nakamura et al., 2018*). To determine whether this is also true for NBR1-decorated chloroplasts, we co-expressed mCherry-NBR1 with the tonoplast marker YFP-VAMP711 (*Geldner et al., 2009*) in *nbr1* seedlings. After HL exposure, mCherry-NBR1-positive chloroplasts associated with deep tonoplast invaginations (*Figure 4A*), which led to their vacuolar internalization by microautophagy, in a process topologically analogous to that previously described for ATG8-decorated chloroplasts (*Izumi et al., 2017*). Similarly, we were able to detect NBR1-positive chloroplasts inside vacuoles of the mCherry-NBR1 seedlings stained with the vacuolar dye BCECF (*Scheuring et al., 2015*) 24 hr after HL exposure (*Figure 4B and C*). Surprisingly, NBR1-decorated chloroplasts were also seen inside vacuoles of *atg7* seedlings (*Figure 4B and C*).

Collectively, these results are consistent with NBR1 associating with chloroplasts targeted for vacuolar degradation through ATG7-independent microautophagy. In addition, the higher number of NBR1-positive photodamaged chloroplasts in *atg7* seedlings (*Figure 2*) did not seem to arise from a failure to deliver these chloroplasts to the vacuole but more likely to the higher accumulation of photodamaged chloroplasts in the *atg7* mutant.

## Impaired remodeling of chloroplasts in *atg7* and *nbr1* mutants

If NBR1 is critical for targeting photodamaged chloroplasts to the vacuole, we reasoned that *nbr1* mutations would reduce the loss of chloroplasts after HL, as it has been shown for the *atg5* and *atg7* mutants (*Izumi et al., 2017*). To test this scenario, we expressed RECA-GFP, a stromal marker bearing the transit peptide of *Arabidopsis* RECA fused to GFP (*Köhler et al., 1997*; *Spitzer et al., 2015*), and imaged the cotyledon sub-adaxial epidermal mesophyll layer from 8-day-old seedlings by confocal microscopy. We found that, under LL conditions *atg7* but not *nbr1* mutant seedlings had a higher

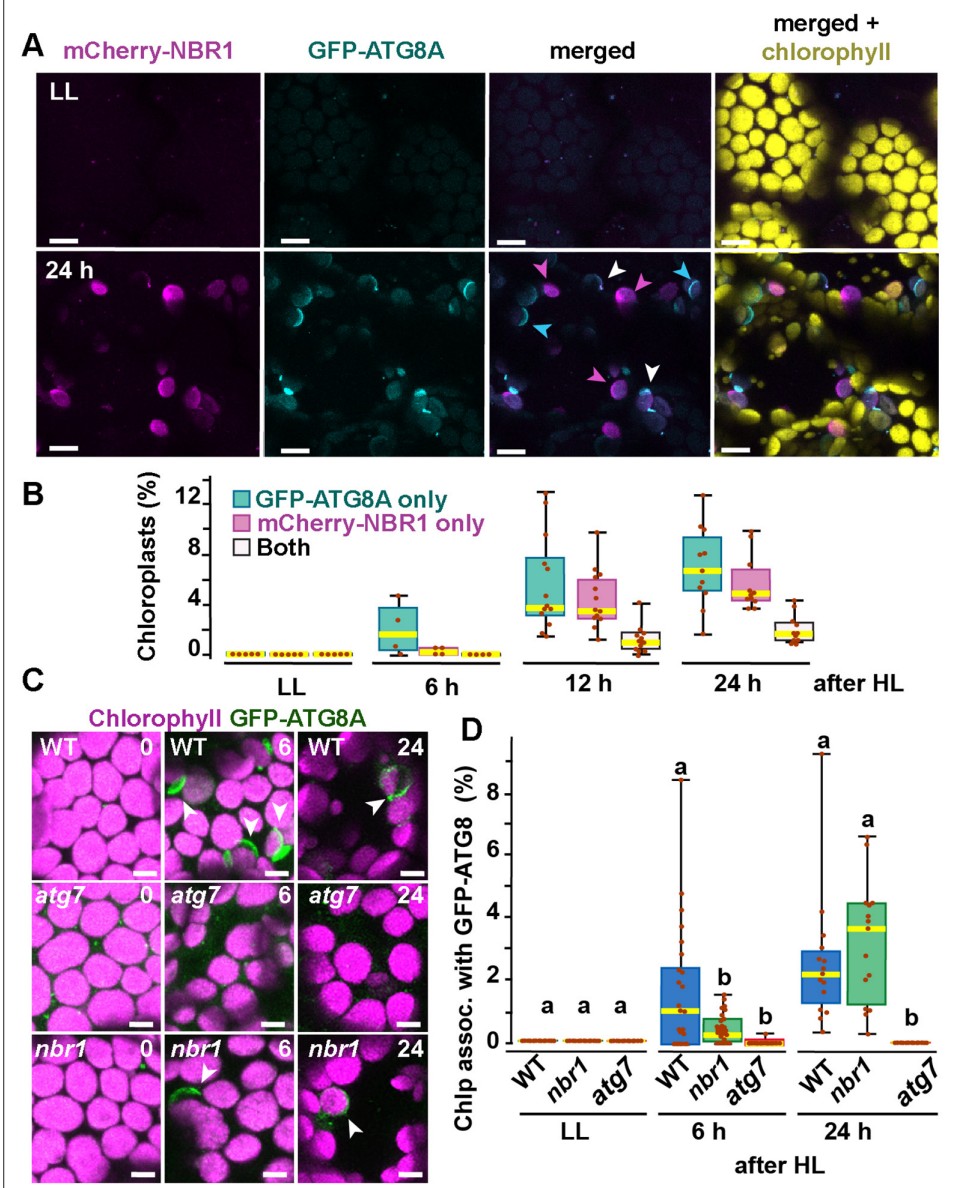

**Figure 3.** Recruitment of NBR1 and ATG8A to photodamaged chloroplasts. (**A**) Confocal imaging of cotyledon mesophyll cells from 8-day-old seedlings expressing mCherry-NBR1 and GFP-ATG8A under LL (top) and at 24 hr after HL exposure (bottom). Magenta, cyan, and white arrowheads indicate chloroplasts associated with mCherry-NBR1, GFP-ATG8, or both, respectively. (**B**) Box and whisker plots showing the percentage of chloroplasts associated with GFP-ATG8A only (cyan), mCherry-NBR1 only (magenta), or both (white) under LL and during recovery after HL exposure. Between 4 and 13 regions containing 20–30 chloroplasts from two seedlings for each time point/treatment were used for this analysis. (**C**) Confocal imaging of GFP-ATG8A in cotyledons of 8-day-old wild-type Col-0 (WT), *atg7*, and *nbr1* seedlings grown under LL, and 6 and 24 hr after HL treatment. Arrowheads indicate chloroplasts associated with GFP-ATG8A. (**D**) Box and whisker plot displaying the percentage of chloroplast associated with GFP-ATG8A in different genotypes, under LL and recovery after HL. Different letters on the graph indicate significant difference (p<0.05) calculated by one-way ANOVA followed by Tukey's test. Box and whisker plots in B and D show the variation in data through quartiles; the middle line indicates the median and whiskers show the upper and lower fences. Between 4 and 13 regions containing 20–30 chloroplasts from two seedlings at each time point/treatment were used for this analysis. Scale bars: 10 µm in A and C.

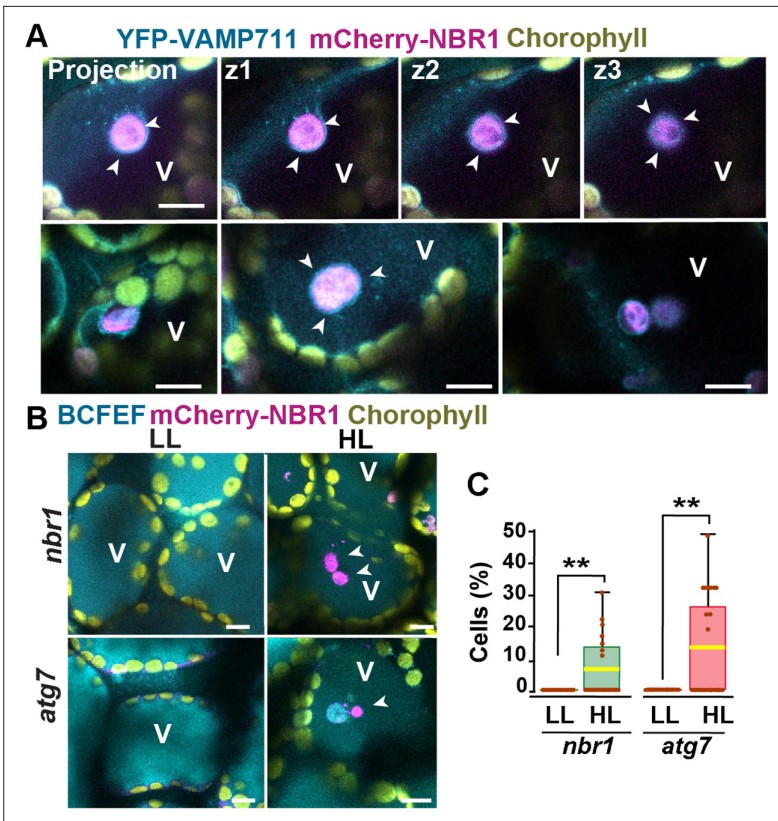

**Figure 4.** Vacuolar delivery of NBR1-positive chloroplast into the vacuole. (**A**) Projection of three confocal images (z1–z3) and several other confocal images of cotyledon mesophyll cells from 1-week-old, wild-type seedlings expressing the tonoplast marker YFP-VAMP711 and mCherry-NBR1, 24 hr after HL exposure. Chloroplast labeled by mCherry-NBR1 were surrounded by the tonoplast (arrowheads) and internalized into the vacuole (V) through microautophagy. (**B**) Confocal images of *nbr1* and *atg7* cotyledon mesophyll cells at 24 hr after HL exposure and stained with the vacuolar dye BCECF. Note the mCherry-NBR1-labeled chloroplasts inside the vacuoles. (**C**) Box and whisker plot displaying the percentage of cells containing mCherry-NBR1-labeled chloroplasts inside their vacuoles. Boxes show the variation in data through quartiles; the middle line indicates the median and whiskers show the upper and lower fences. A t-test was used to compare values between LL and recovery after HL ** indicate p<0.01. Between 13 and 27 regions containing 7-10 cells from at least two seedlings of each treatment and genotype were used for this analysis. Scale bars: 10 µm in A and B.

density of chloroplasts compared to wild type (*Figure 5A and B*). Twenty four hours after HL, there was a reduction in chloroplast density in all three genotypes but the decrease was less pronounced in *atg7* and *nbr1* (17% and 16% reduction, respectively) compared to the wild-type control (25% reduction; *Figure 5A and B*), consistent with impaired turnover of chloroplasts in both *atg7* and *nbr1* mutants.

To examine whether chloroplast size also changed upon HL exposure, we measured the area of both RECA-GFP signal (stroma) and chlorophyll autofluorescence (thylakoids) in individual chloroplasts. Overall, there was a decrease in both RECA-GFP and chlorophyll area of individual chloroplasts 24 hr after HL exposure in the three genotypes. However, whereas the *nbr1* and wild type RECA-GFP-decorated chloroplasts were similar in size under LL, the decrease in RECA-GFP area upon HL treatment was slightly more abrupt in *nbr1* (19% reduction) than in control cotyledons (14% reduction; *Figure 5A and C*). RECA-GFP-decorated *atg7* chloroplasts were smaller than those in control and *nbr1* cotyledons and showed a small (6%) reduction in area upon HL treatment (*Figure 5A and C*). Chlorophyll areas were smaller in *nbr1* and *atg7* chloroplasts compared to the wild-type control even under LL conditions, and underwent an attenuated reduction after HL exposure (21% and 18% in *nbr1* and *atg7*, respectively), compared to control chloroplasts (36% reduction; *Figure 5A and D*). These results demonstrate that although both *atg7* and *nbr1* retained more chloroplasts than control plants

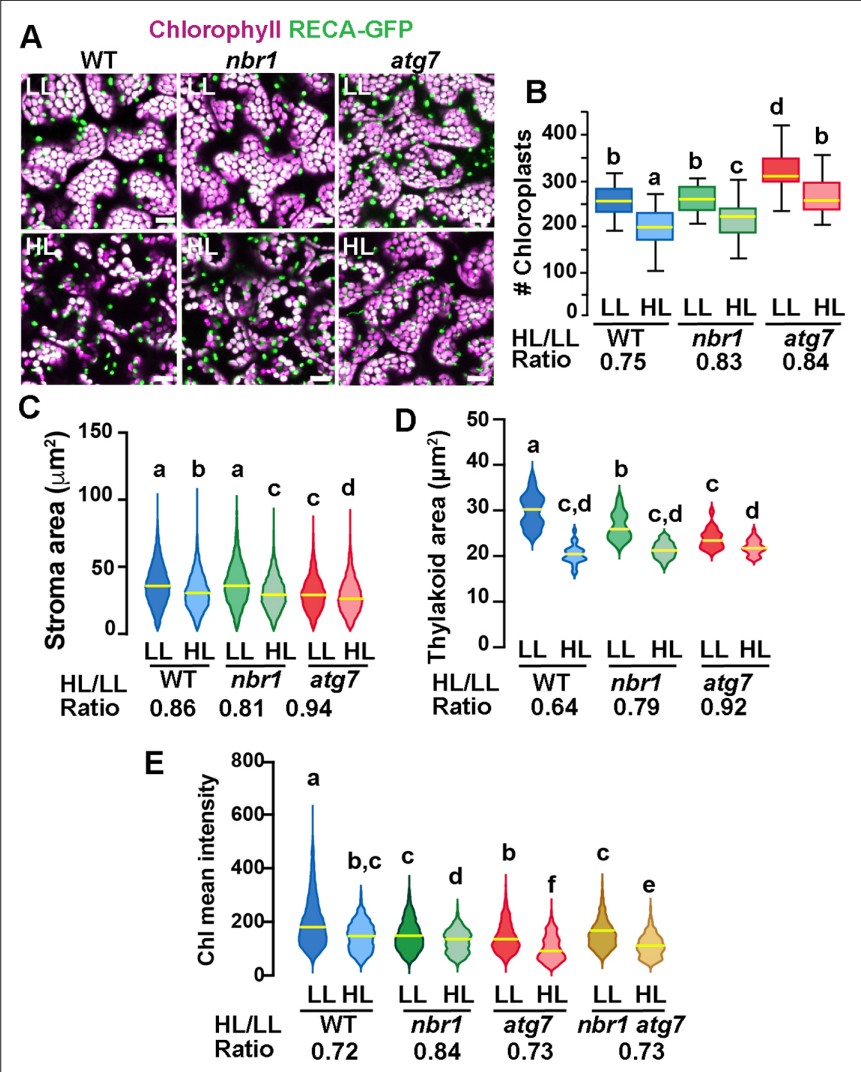

**Figure 5.** Chloroplast remodeling after HL exposure. (**A**) Projections of 20 confocal images along a z-stack taken from the adaxial side of cotyledon mesophyll cells from 8-day-old wild-type (WT), *atg7*, and *nbr1* seedlings expressing RECA-GFP. Seedlings were grown under LL, exposed to HL for 2 hr and let recover for 24 hr. (**B**) Chloroplast density in adaxial-facing mesophyll cells (number of chloroplasts per 2.16 µm²) based on confocal images. At least 20 randomly selected areas from 6 to 9 cotyledons were considered in this analysis. Boxes show the variation in data through quartiles; the middle line indicates the median and whiskers show the upper and lower fences. (**C**) Stroma area (µm²) as measured by the RECA-GFP fluorescence signal per individual chloroplast imaged by confocal microscopy. Lines in violin plots indicate median values. At least 25 individual chloroplasts were measured for each genotype and condition. (**D**) Thylakoid area (µm²) as measured by chlorophyll fluorescence signal area per individual chloroplast. Line in violin plots indicate median values. At least 5000 individual chloroplasts were measured for each genotype and condition. (**E**) Chlorophyll mean intensities measured in individual chloroplasts by multiphoton imaging. Between 1300 and 2600 individual chloroplasts were measured for each genotype and condition. Lines in violin plot indicate median values. In B to E, the HL/LL ratio was calculated by dividing the average value from HL-treated plants by the average value of the plants grown under LL. Different letters denote significant differences from each other based on two-way ANOVA followed by Tukey's test ($p < 0.05$). Scale bars: 10 µm in A.

after HL exposure (*Figure 4E*), the remaining chloroplasts in the mutants were smaller, both in stroma and thylakoid areas.

These unique chloroplast dynamics in *nbr1*, *atg7*, and wild-type plants suggested that although ATG7 and NBR1 are both important for chloroplast turnover, they control different aspects of chloroplast remodeling/turnover after HL radiation. To further understand how chloroplasts are differentially

affected, we used multiphoton imaging to excite and measure chlorophyll mean intensities under LL and 24 hr after HL exposure in *nbr1* and *atg7* seedlings, together with a previously characterized *nbr1-2 atg7-2* double mutant (*Jung et al., 2020*). Compared to controls, mean chlorophyll fluorescence intensity in all three mutants was weaker than wild type under LL conditions. This intensity decreased approximately 27–28% in wild type and the *atg7* mutant after HL (based on ratio of chlorophyll mean intensities between HL and LL values), but only 16% in the *nbr1* mutant (*Figure 5E*). Unexpectedly, the mean chlorophyll intensity values from *nbr1 atg7* mutant under LL and after HL treatment were intermediates between those from the single *nbr1* and *atg7* mutant seedlings. These results showed that mutations in both *NBR1* and *ATG7* affect either chlorophyll abundance and/or chlorophyll photochemical properties even under LL conditions.

## Proteome profiling supports NBR1- and ATG7-dependent pathways for clearing photodamaged chloroplasts

To further understand the function of NBR1 and ATG7 in chloroplast remodeling and turnover, we analyzed by tandem mass spectrometry (MS/MS) the total proteome of 1-week-old *atg7*, *nbr1*, *nbr1 atg7* double mutant, and wild-type seedlings grown under LL and at 24 hr after HL treatment (*Figure 6*, *Figure 6—figure supplements 1 and 2*, *Supplementary file 1a-n*).

In total, 4,975 proteins were identified in the 48 datasets (three biologicals replicates of the four genotypes exposed to the two different light treatments, each analyzed as two technical replicates), of which 3806 proteins were selected after filtration of our criteria for further analysis (*Supplementary file 1*; *Figure 6—figure supplement 1A*). Principal component analyses (PCA) of the datasets showed that samples clustered by genotype under both LL and HL (*Figure 6—figure supplement 1B*). The proteomic profiles from the *nbr1 atg7* double mutant, either under LL or HL, was place by PCA close to those of *atg7* mutant plants, suggesting that the *atg7* mutation has a dominant effect on the proteome of the *nbr1atg7* double mutant.

In terms of relative protein abundance, the HL treatment caused significant changes in the total proteome of all four genotypes. Approximately 4.5%, 17%, 8.5%, and 6% of the total identified proteins showed significant abundance changes in the wild type, *atg7*, *nbr1*, and *nbr1 atg7* plants, respectively (*Figure 6—figure supplement 1A*, *Supplementary file 1d-g*). For those proteins showing significant changes upon HL exposure (fold change >1.5 or<1.5), we found strong correlations between all three mutants (*Figure 6—figure supplement 1C*, *Supplementary file 1h-n*), suggesting that NBR1 and ATG7 have overlapping roles in regulating global proteome homeostasis after HL exposure.

Based on Gene Otology (GO) term analysis, over 1100 of the total proteins identified by MS/MS could be assigned to plastid-type compartments, thus interrogating most, if not all, functions associated with this compartment. When analyzing this collection separately, we found that plastid proteins were well represented among those with decreased abundance upon HL exposure in wild type and the mutants (*Figure 6—figure supplement 1D*), which was consistent with the overall reduction in chloroplast density and sizes seen in both backgrounds upon HL exposure (*Figure 5*). However, in the three mutants, but not in the wild type, proteins with increased abundance were also found associated with plastids and organelles (*Figure 6—figure supplement 1D*), consistent with impaired organelle turnover caused by the *atg7* and *nbr1* mutations. These results implied that whereas all four genotypes showed an overall reduction in the abundance of some chloroplast proteins upon exposure to photodamaging radiation, only the *atg7*, *nbr1*, and *atg7 nbr1* mutants showed a significant accumulation of a subset of chloroplast proteins, which we hypothesized was caused by a lack of autophagic clearance.

The better understand the changes in the chloroplast proteome, we analyzed separately chloroplast proteins in the four genotypes. The wild-type plants showed an overall decrease in chloroplast proteins after HL treatment (2.5% of total chloroplast proteins were less abundant and 1% were more abundant compared to LL conditions) (*Figure 6A*). By contrast, the three mutants showed a more pronounced increase in chloroplast protein abundance (*Figure 6A*) compared to wild type. For example, in the *atg7* mutant, 9% were more abundant after HL exposure and only 3.5% were less abundant, while in *nbr1* mutant, approximately the same number of chloroplast proteins (3% of the total chloroplast proteins) showed significant increase or decrease in abundance after HL. The chloroplast proteins accumulating in the mutants localized to chloroplast envelopes, stroma, and thylakoids (*Figure 6—figure supplement 2*), indicating that the *atg7* and *nbr1* mutations affects the homeostasis

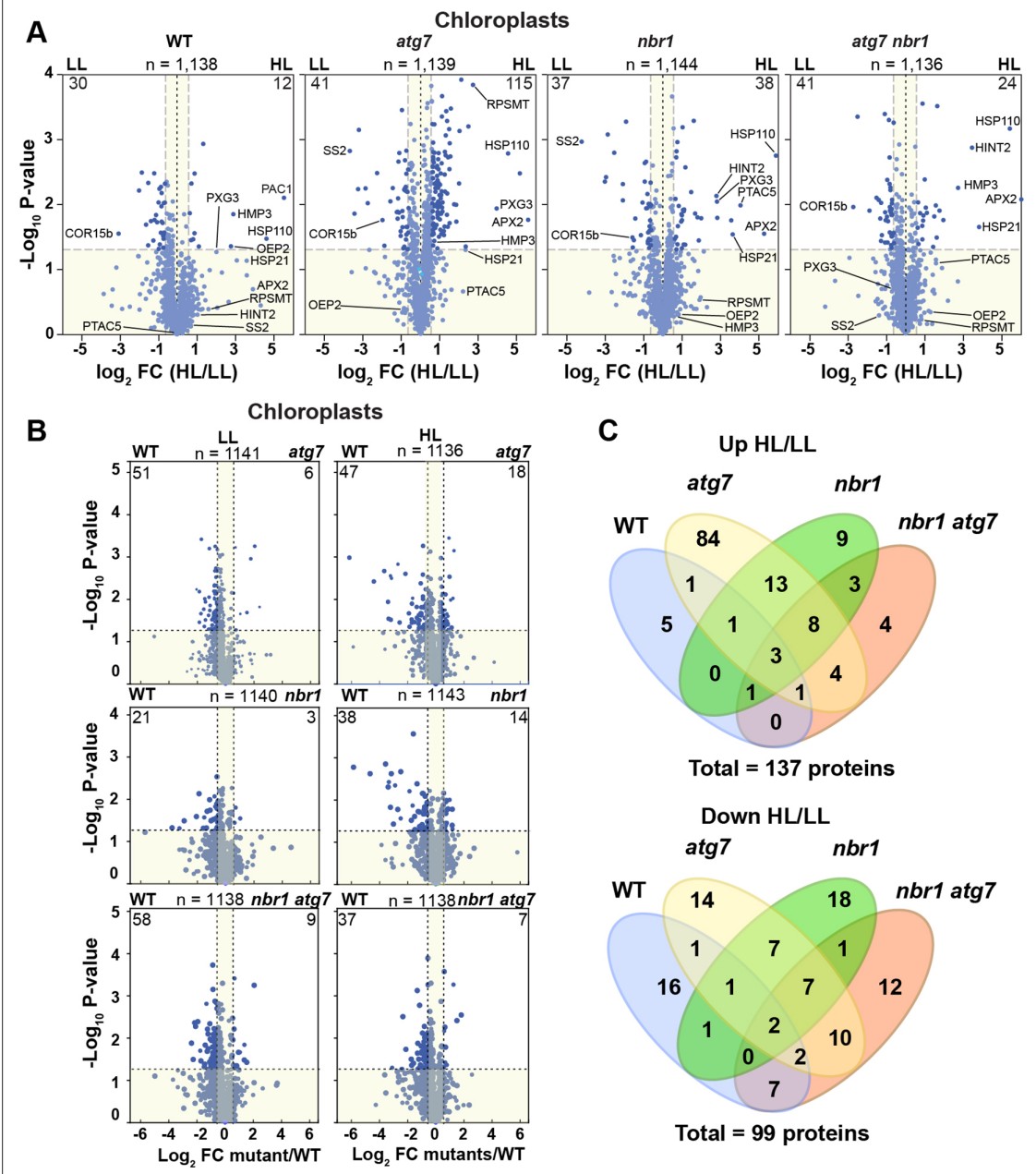

**Figure 6.** Chloroplast proteome analysis by liquid chromatography-tandem mass spectrometry (LC-MS/MS). (**A**) Volcano plots showing the changes in the relative abundance of chloroplast proteins under LL or HL, in wild type (WT) and mutants. The number on the top of each plot indicates the total number of detected proteins assigned by GO to chloroplasts. Several representative proteins are labeled in each plot. The lighter blue points identify proteins with insignificant changes, while the darker blue points identify those that meet a significance threshold of FC > 1.5 or –1.5 and p-value < 0.05. The numbers at the left and right corners of each plot indicate the less or more abundant proteins, respectively. (**B**) Volcano plots as in (**A**) showing the changes in the relative abundance of chloroplast proteins in mutants relative to WT, either under LL or HL. (**C**) Venn diagrams depicting the overlap among chloroplast proteins changing abundance between HL and LL conditions in mutants and WT plants.

The online version of this article includes the following figure supplement(s) for figure 6:

**Figure supplement 1.** Proteome analysis by liquid chromatography-tandem mass spectrometry (LC-MS/MS).

**Figure supplement 2.** Proteome analysis by liquid chromatography-tandem mass spectrometry (LC-MS/MS) of chloroplast proteins localized to envelopes, stroma, and thylakoid membranes.

of all chloroplast subcompartments when plants are exposed to HL. Interestingly, when compared to the wild type, it became evident that both under LL and HL conditions, the three mutants contain less chloroplast proteins (*Figure 6B*), suggesting that chloroplast homeostasis is regulated by both ATG7 and NBR1, even under non-photodamaging conditions.

We then compared the overlap of more and less abundant proteins in the four genotypes in response to HL. We found that 66% (25 of 38) of the chloroplast proteins significantly enriched in *nbr1* were also more abundant in *atg7*, whereas only 22% (25 of 115) of the more abundant chloroplast proteins in *atg7* were also enriched in *nbr1* (*Figure 6C*). Of the 24 enriched proteins in the *nbr1 atg7* double mutant, 16 proteins (67%) were shared with either *atg7, nbr1,* or both. For chloroplast proteins with decreased abundance upon HL exposure, the overlap between the single mutants was 38% (17 of 44 proteins) for *atg7* and 46% (17 of 37 proteins) for *nbr1* (*Figure 6C*). Of the 41 proteins decreasing in abundance in *nbr1 atg7* plants, 21 proteins (51%) also showed reduced abundance in one or both single mutants. These results suggest that both NBR1 and ATG7 controls the degradation and overall homeostasis of chloroplasts after HL damage, but their functions are only partially overlapping.

## Contributions of NBR1 domains to chloroplast recruitment

To identify the NBR1 domains that help NBR1 associate with photodamaged chloroplasts, we expressed in the *nbr1-1 background* several mutant versions of YFP-NBR1 missing key domain functions, such as NBR1-mPB1 with a point mutation (K11A) in the PB1 domain (*Figure 7A and B*) that disrupts NBR1 oligomerization (*Hafrén et al., 2017*), NBR1-mAIM with two point mutations in the AIM domain (W661A, I664A) that block interaction with ATG8 (*Svenning et al., 2011*; *Hafrén et al., 2017*), and NBR1-ΔUBA2 missing the UBA2 domain (*Figure 7A and B*) and therefore, unable to bind ubiquitin (*Svenning et al., 2011*; *Hafrén et al., 2018*). All fluorescent NBR1 fusion proteins remained cytosolic under LL condition (*Figure 7B and C*). After HL exposure, YFP-NBR1 associated with photodamaged chloroplasts as expected, either forming coats (average ~3% of chloroplasts; n=28 fields) or localizing inside a small fraction of chloroplasts (average ~0.5% of chloroplasts; n=28 fields) (*Figure 7B and C*). YFP-NBR1mPB also localized to photodamaged chloroplasts but almost exclusively to their interior (*Figure 7B and C*). Thus, although the total percentages of chloroplasts labeled by YFP-NBR1 and YFP-NBR1mPB were similar (*Figure 7C*), YFP-NBR1 mainly coated the chloroplast surface whereas most of YFP-NBR1mPB was located to the chloroplast lumen. Just like YFP-NBR1, YFP-NBR1mAIM was mostly detected as chloroplast coats (*Figure 7B and C*). Interestingly, YFP-NBR1ΔUBA2 failed to associate with chloroplasts after HL exposure (*Figure 7B and C*). The expression of the same set of NBR1 mutated proteins resulted in a similar pattern of chloroplast association in the *atg7 nbr1* seedlings exposed to HL (*Figure 7—figure supplement 1*). Thus, these results suggest the UBA2 domain is required for NBR1 to associate with chloroplasts, whereas the PB1 domain negatively regulates NBR1 intra-chloroplast localization and/or promotes degradation of NBR1-filled chloroplasts.

## The E3 ligases PUB4 and SP1 are not required for NBR1 association with photodamaged chloroplasts

Because the UBA2 ubiquitin-binding domain of NBR1 is critical for chloroplast binding upon HL treatment, we expressed NBR1-GFP in mutants lacking the E3 ligases PUB4 and SP1, which have been shown to ubiquitylate chloroplast envelope proteins after HL stress as part of the CHLORAD pathway (*Ling et al., 2012*; *Woodson et al., 2015*). NBR1-GFP localized to photodamaged chloroplasts in *pub4-2* and *sp1-2* single and double mutants during recovery after HL (*Figure 8*), suggesting that at least these two E3 ligases are not critical for NBR1 association with photodamaged chloroplasts.

## Fully functional TIC-TOC complexes are not required for NBR1 internalization into chloroplasts

To test whether NBR1 is translocated into the chloroplast stroma *via* the TIC-TOC complexes, we expressed NBR1-GFP in the transcript-null *toc132-2* mutant, which is defective in the import of a subset of chloroplast proteins (*Kubis et al., 2004*). The *toc132-2* mutation did not affect the association of NBR1-GFP with chloroplasts or its localization into the chloroplast lumen (*Figure 9A and B*).

As the *toc132-2* mutation affects the translocation of only some but not all chloroplast proteins into the stroma (*Kubis et al., 2004*), we also tested NBR1 localization in the transcript-knockout *tic40-4* mutant, which is severely deficient in chloroplast protein import and consistently develops

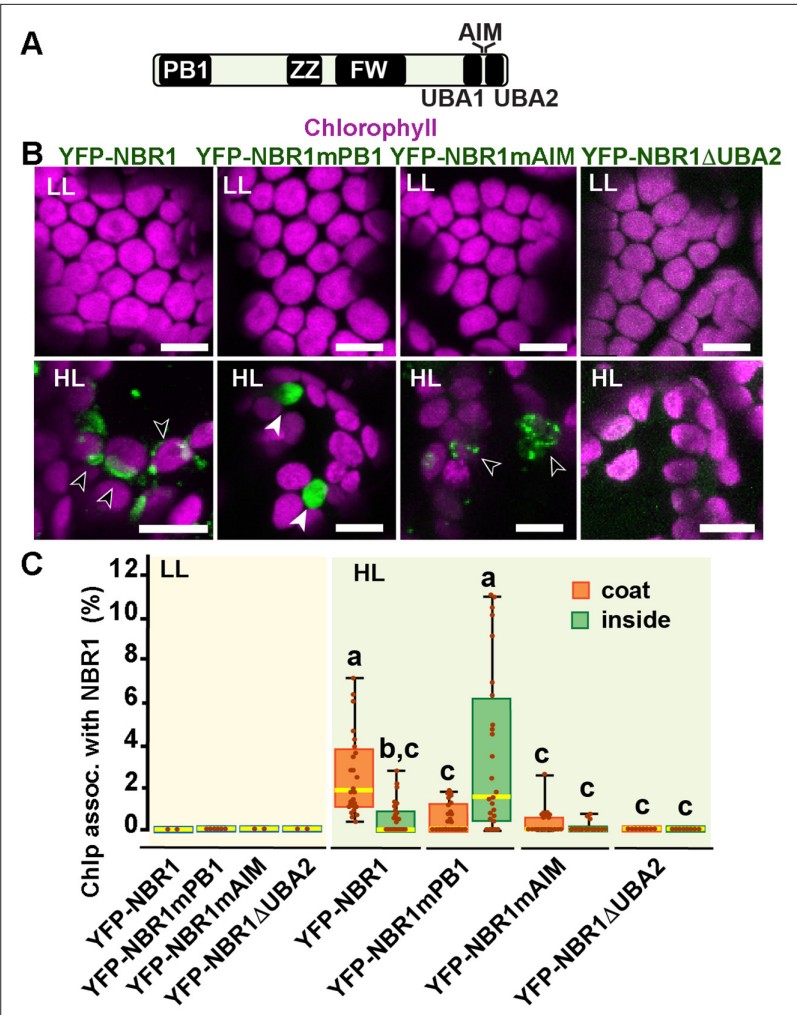

**Figure 7.** NBR1 domains have distinct roles in recruiting NBR1 to chloroplasts after HL treatment. (**A**) Diagram of the *Arabidopsis* NBR1 protein and its domains. FW, Four-Tryptophan domain; PB1, Phox and Bem1p domain; ZZ, ZZ-type zinc finger domain; UBA1 and UBA2, ubiquitin-associated domains; AIM, ATG8-interacting motif. (**B**) Confocal imaging of NBR1 mutated proteins fused to YFP expressed in 8-day-old *nbr1* seedlings grown under LL (top) or at 24 hr after HL exposure (bottom). Hollow arrowheads and filled arrowheads indicate YFP-NBR1 coating chloroplasts and inside chloroplasts, respectively. (**C**) Box and whisker plots show the percentages of chloroplast associated with the YFP-labeled mutated NBR1 proteins, localized to either coats (orange) or inside chloroplasts (green). Boxes show the variation in data through quartiles; the middle line indicates the median and whiskers show the upper and lower fences. Different letters denote significant differences from each other based on two-way ANOVA followed by Tukey's test (p<0.05). Between 2 and 26 regions containing 20–30 chloroplasts from at least two seedlings for each treatment and NBR1 construct were used for this analysis. Scale bars: 10 μm in B.

The online version of this article includes the following figure supplement(s) for figure 7:

**Figure supplement 1.** NBR1 domains in NBR1 recruitment to chloroplasts in *nbr1 atg7* double mutant cotyledons after HL treatment.

structurally abnormal chloroplasts (***Kovacheva et al., 2005***). We isolated protoplasts from 3-week-old *tic40-4* and wild-type seedlings and transfected them with the *pUBN-NBR1mPB1* vector, since the NBR1mPB1 protein is internalized into photodamaged chloroplasts at a higher rate than the wild-type NBR1 protein (***Figure 7C***). We exposed the transfected protoplasts to HL for 2 hr and imaged them 12 hr later. We found that YFP-NBR1mPB1 successfully internalized into photodamaged chloroplasts in *tic40-4* mutant protoplasts. In fact, we detected a larger proportion of chloroplasts with internal YFP-NBR1mPB1 signal in the *tic40-4* compared to wild-type protoplasts (***Figure 9***. C, D). From this,

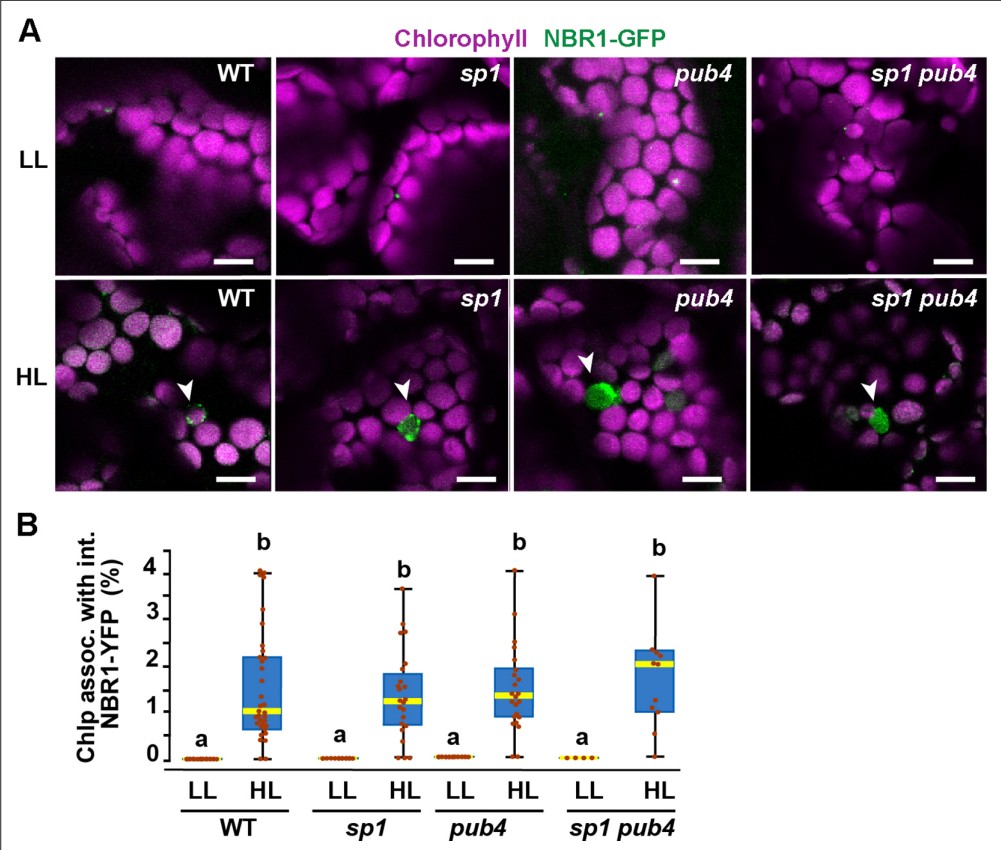

**Figure 8.** NBR1 association with chloroplasts in mutants lacking SP1 and PUB4 E3 ligases. (**A**) Confocal imaging of NBR1-GFP in 8-day-old wild type (Col-0), *sp1*, *pub4*, and *sp1 pub4* seedlings under LL and 24 hr after HL exposure. Arrowheads indicate chloroplasts decorated with NBR1-GFP. (**B**) Box and whisker plots show the percentage of chloroplast associated with NBR1-GFP under LL and 24 hr after HL treatment. Boxes show the variation in data through quartiles; the middle line indicates the median and whiskers show the upper and lower fences. Different letters denote significant differences from each other based on two-way ANOVA followed by Tukey's test (p<0.05). Between 4 and 34 regions containing 20–30 chloroplasts from at least two seedlings of each treatment and genotype were used for this analysis. Scale bar: 10 µm in A.

we concluded that the TIC-TOC complex is not required for the internalization of NBR1 into chloroplasts after HL exposure.

## Discussion

Here, we present evidence that the selective autophagy receptor NBR1 is recruited to photodamaged chloroplasts, mediating their clearance by a microautophagy-like mechanism that is independent of the canonical ATG machinery (*Figures 1–4*). Upon photoradiation damage, NBR1 first becomes associated with the chloroplast surface to be later internalized into the chloroplast stroma (*Figure 1E*). The association of NBR1 with chloroplasts requires its ubiquitin-binding UBA2 domain whereas NBR1 internalization into the chloroplast stroma is negatively regulated by its self-polymerization PB1 domain (*Figure 7*). The relocation of NBR1 into the chloroplast stroma does not rely on a functional TIC-TOC complex (*Figure 9*). We propose that the rupture of the outer and inner envelopes in photodamaged chloroplasts (*Figure 2C*) allows for the diffusion of the ubiquitylation machinery and NBR1 from the cytosol into the chloroplast lumen, promoting ubiquitylation of both stroma and thylakoid proteins and their subsequent binding to NBR1 for vacuolar degradation via microautophagy (*Figure 9E*).

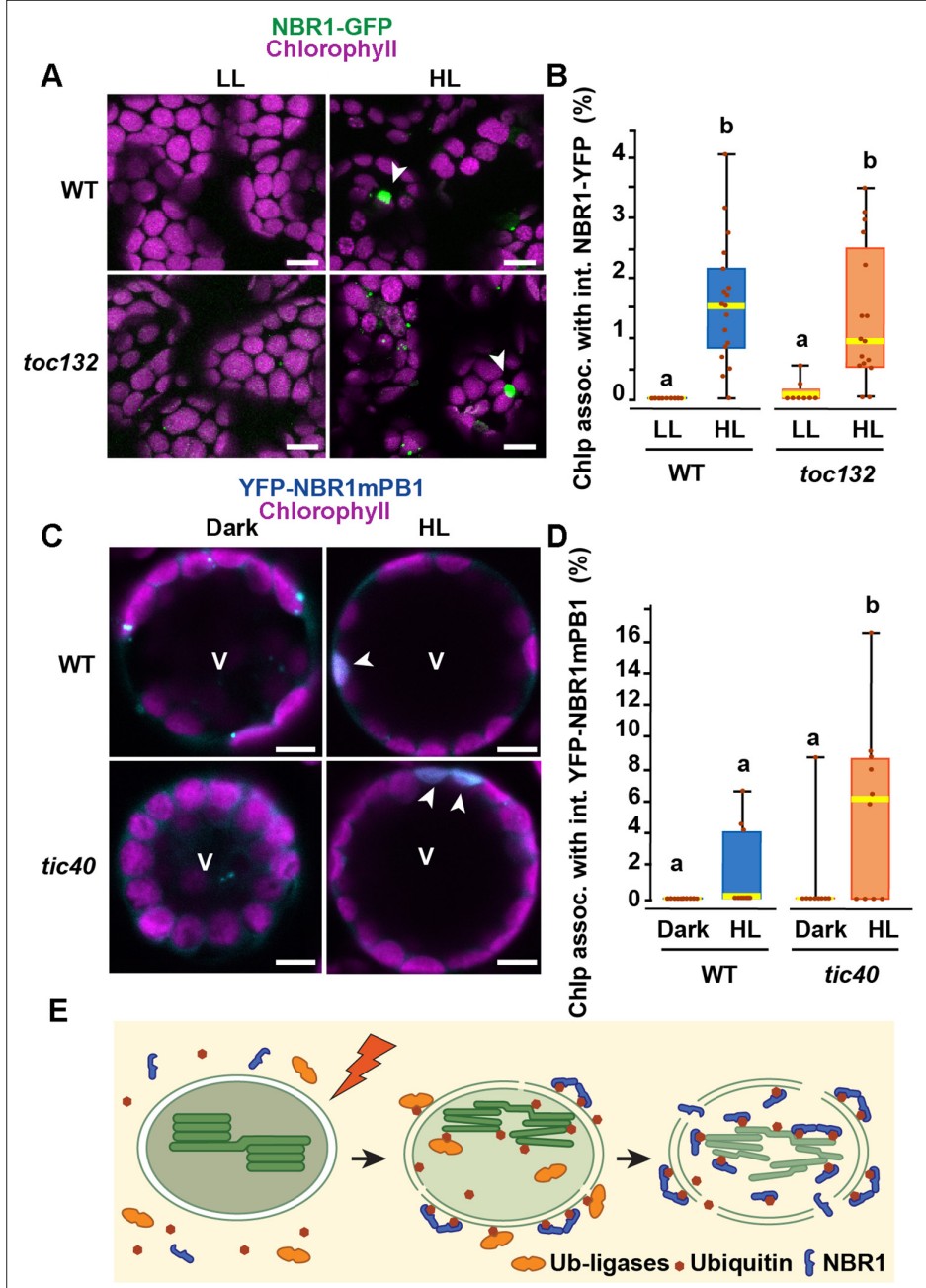

**Figure 9.** The TIC-TOC translocon is not required for the internalization of NBR1 into photodamaged chloroplasts. (**A**) Confocal imaging of NBR1-GFP in wild-type Col-0 (WT) and *toc132* cotyledon mesophyll cells from 1-week-old seedlings grown under LL or at 24 hr after HL exposure. Arrowheads indicate chloroplasts with internal NBR1-GFP signal. (**B**) Box and whisker plot displaying the percentages of chloroplasts associated with NBR1-GFP signal in WT and *toc132* mutant mesophyll cells under LL or at 24 hr after HL exposure. Between 8 and 18 regions containing 20–30 chloroplasts from at least 3 cotyledons were used for quantification. (**C**) Protoplasts from 3-week-old wild-type Col-0 (WT) and *tic40-4* expressing YFP-NBR1mPB1. Protoplasts were left in the dark or exposed to HL for 2 hr and imaged 12 hr later. Arrowheads indicate chloroplasts with internal YFP-mPB1-NBR1 signal. V, vacuole. (**D**) Box and whisker plot displaying the percentages of wild type and *tic40-4* chloroplasts associated with YFP-mPB1-NBR1 signal in WT and *tic40-4* mutant protoplasts kept in dark conditions or exposed to HL and left to recover for 12 hr. Between 9 and 10 protoplasts of each genotype and condition were used for quantification. In B and D, boxes show the variation in data through quartiles; the middle line indicates the median and whiskers show the upper and lower fences. Different letters denote significant differences from each other based on two-way ANOVA followed by Tukey's test (p < 0.05). (**E**) Diagram summarizing a proposed mechanism for NBR1 association

*Figure 9 continued on next page*

*Figure 9 continued*

with photodamaged chloroplasts. HL exposure induces the breakdown of the chloroplast envelopes allowing the cytosolic ubiquitylation machinery to reach the stroma and thylakoids of photodamaged chloroplasts. As stromal and thylakoidal proteins become ubiquitylated, NBR1 diffuses into damaged chloroplasts and bind ubiquitylated proteins through its UBA2 domain. NBR1-decorated photodamaged chloroplasts are then delivered to the vacuole by microautophagy independently of ATG7. Scale bars: 5 µm in A and C.

## NBR1 as a ubiquitin-binding chlorophagy receptor

NBR1 is a well-known aggrephagy receptor that recognizes and sorts ubiquitylated cargo for vacuolar clearance (*Rasmussen et al., 2022*). In plants, the formation of ubiquitylated cargo aggregates by NBR1 depends on it self-oligomerizing PB1 domain and its ubiquitin-binding capacity through the UBA2 domain (*Svenning et al., 2011*; *Zientara-Rytter and Sirko, 2014*). The AIM domain of NBR1 mediates its interaction with ATG8 and its sequestration into autophagosomes for vacuolar degradation (*Svenning et al., 2011*). Our studies found that, upon high photoradiation exposure, NBR1 associates with a population of photodamaged chloroplasts via a process dependent on its UBA2 domain, which then enables the association of NBR1 with both the surface of the chloroplast and its lumen (stroma and thylakoids).

A simple scenario based on past studies is that NBR1 binds ubiquitylated substrates of the E3 ligases PUB4 and SP1, which ubiquitylate chloroplast envelope proteins as part of the CHLORAD pathway (*Ling et al., 2012*; *Woodson et al., 2015*). However, we found that a mutant lacking both PUB4 and SP1 activity showed normal recruitment of NBR1 to photodamaged chloroplasts (*Figure 8*). Recent reports have also shown that most chloroplast proteins, including those localized to stroma and thylakoids are ubiquitylated for subsequent break down by the proteasomes (*Li et al., 2022*; *Sun et al., 2022b*), but how such ubiquitylation might occur inside chloroplasts was unresolved. As rupture of the chloroplast envelope membranes is a known consequence of damaging photoradiation (*Nakamura et al., 2018*), it is important to note our observations that NBR1 heavily decorates the surface, stroma, and thylakoids of photodamaged chloroplasts with structurally disrupted envelopes (*Figure 2E and F*). Consequently, we hypothesize that the loss of envelope structural integrity allows access of the cytosolic ubiquitylation machinery to the stroma and thylakoids of compromised chloroplast thus directing the massive ubiquitylation of chloroplast proteins for recognition by the NBR1 receptor. Although ubiquitylation of intra-chloroplast proteins has been connected to degradation by the 26 S proteasome through CHLORAD (*Li et al., 2022*; *Sun et al., 2022b*), it is also possible that remaining ubiquitylated chloroplast ghost membranes coated with NBR1 are delivered to the vacuole by microautophagy.

The mPB1 domain of NBR1 is necessary for aggrephagy in plants as it mediates the formation of ubiquitylated cargo accretions (*Svenning et al., 2011*). Here, we show that an NBR1 protein unable to oligomerize is relocated into the chloroplast lumen at a higher frequency than the wild-type NBR1 protein (*Figure 7B and C*). We speculate that monomeric NBR1 proteins diffuses more easily through disrupted envelope membranes to reach the normally inaccessible chloroplast stroma where they bind ubiquitylated chloroplast proteins.

Surprisingly, although NBR1 targets photodamaged chloroplasts for vacuolar clearance, this process requires neither its ATG8-interacting AIM domain nor ATG7, and thus is independent of canonical autophagy. Although microautophagy of ATG8-decorated chloroplasts upon radiation damage requires the core ATG machinery that assembles the ATG8-PE adduct (*Izumi et al., 2017*), microautophagy of chloroplasts damaged by oxidative stress does not (*Lemke et al., 2021*). Thus, as a protein targeting chloroplasts for non-canonical microautophagy, it is possible that NBR1 also mediates the clearance of chloroplast damaged by oxidative stress.

Autophagy defective maize (*atg12*) and *Arabidopsis* (*atg2*, *atg5*, *atg7*, and *atg9*) mutant plants show reduced abundance of chloroplast proteins under different developmental and environmental conditions (*McLoughlin et al., 2018*; *McLoughlin et al., 2020*; *Wijerathna-Yapa et al., 2021*), despite autophagy being a catabolic pathway. This unexpected increase could be attributed to either a lower nutrient availability in autophagy defective lines, which results in lower protein biosynthesis, or the induction of alternative proteolytic pathways to compensate for the lack of autophagy. With the limitation that this study focused on, albeit well characterized, single mutant alleles, in the absence of nutritional deficiency, we also observed a lower abundance of chloroplast proteins for all autophagy

defective lines (*atg7*, *nbr1*, and *nbr1 atg7*) after HL exposure (*Figure 6B*), consistent with either the induction of other proteolytic route(s) and/or a delay in chloroplast protein synthesis and recovery after photoradiation damage. In this context, both *nbr1* and *nbr1 atg7* plants, showed a lessened reduction in chloroplast protein abundance relative to *atg7* (*Figure 6B*). We speculate that all these lines display enhanced degradation of chloroplast proteins, but mutations in *NBR1* dampen this exacerbated catabolic activity that target chloroplasts when autophagy is blocked.

The role of NBR1 in organelle turnover and remodeling does not seem to be general for all organelles as peroxisomal protein abundance was not altered in *nbr1* backgrounds but significantly elevated in the *atg7* mutant (*Figure 6—figure supplement 2*). This also confirms previous reports that, different from animals (*Deosaran et al., 2013*), plants do not seem to employ NBR1 for autophagic removal of peroxisomes (pexophagy) (*Young et al., 2019*; *Jung et al., 2020*).

## The many pathways promoting chloroplast remodeling and degradation

Chloroplast proteostasis is critical for plant survival, which is constantly challenged by daily exposure to damaging reactive oxygen species generated unavoidably by the photosynthetic machinery (*Foyer, 2018*) and by a hypersensitivity of chloroplasts to biotic and abiotic stresses (*Nishimura et al., 2017*; *Song et al., 2021*; *Wang et al., 2023*). A failure to control chloroplast protein turnover is often very deleterious to plants (*Rowland et al., 2022*). Not surprisingly considering the complexity of the organelle and its functions, chloroplast remodeling and turnover are intricate processes that integrates multiple likely redundant or partially redundant pathways. Besides chloroplast proteases that can locally degrade proteins inside chloroplasts (*Nishimura et al., 2017*; *Rowland et al., 2022*), several autophagy and non-autophagic pathways mediate vacuolar clearance of chloroplast components (*Otegui, 2018*; *Kikuchi et al., 2020*; *Rowland et al., 2022*). At least three flavors of ATG8-dependent piecemeal autophagy of chloroplasts have been characterized: Rubisco-containing bodies (*Chiba et al., 2003*; *Ishida et al., 2008*; *Spitzer et al., 2015*), ATG8-INTERACTING PROTEIN 1 bodies (*Michaeli et al., 2014*), and small starch-like granule bodies (*Wang et al., 2013*). In addition, microautophagy of whole damaged chloroplasts occurs through at least two pathways, one dependent and the other independent of canonical autophagy (*Izumi et al., 2017*; *Lemke et al., 2021*). For the latter pathway, we provide evidence for a novel microautophagic route that requires NBR1 but not ATG8 lipidation.

How exactly all these pathways coordinate the remodeling and degradation of damaged chloroplasts is unclear. Upon HL exposure, we observed chloroplasts associated with either ATG8 and NBR1 as organelle cargo for canonical autophagy-dependent and independent microautophagy, respectively (*Figure 3*). Only a very low proportion of chloroplasts were coated with both ATG8 and NBR1, supporting the notion that there are two separate microautophagy pathways for chloroplast clearance. However, we noticed a higher proportion of NBR1-decorated chloroplasts in HL-exposed *atg7* mutant seedlings compared to controls (*Figures 1F and 4C*). Although this could be due to the increased levels of NBR1 protein in the *atg7* mutant (Fig, 2 G; *Jung et al., 2020*), the higher frequency of photodamaged chloroplasts observed in *atg7* (*Figure 2D*), supports the idea that photodamaged chloroplasts that are not successfully repaired or degraded by canonical autophagy, become substrates of an NBR1-mediated route. Interestingly, the *tic40-4* mutant, which contains structurally abnormal chloroplasts (*Kovacheva et al., 2005*), also shows increased association of chloroplasts with NBR1 upon HL exposure (*Figure 9C*), consistent with more widespread photodamage in the *tic40-4* chloroplasts, which in turn results in more chloroplasts being targeted by NBR1.

We had anticipated that an *nbr1 atg7* double mutant exposed to HL would show more pronounced defects in chloroplast homeostasis after HL exposure than the single mutants if the ATG8- and NBR1-mediated microautophagy pathways were both disrupted. However, a more drastic phenotypic alteration as compared to those seen in the single mutants was not seen in terms of both chlorophyll mean intensities and chloroplast proteome profiles, and instead, the mutant behaved either as intermediate between the two single mutants or more similarly to the *atg7* single mutant. This observation suggests that canonical autophagy controls the main pathway for clearance of photodamaged chloroplasts, whereas NBR1 targets a relatively small population of chloroplasts and chloroplast proteins that fail to be degraded via either CHLORAD or canonical autophagy.

# Materials and methods

**Key resources table**

| Reagent type (species) or resource | Designation | Source or reference | Identifiers | Additional information |
|---|---|---|---|---|
| Gene (*Arabidopsis thaliana*) | NBR1 | | AT4G24690 | |
| Gene (*Arabidopsis thaliana*) | ATG7 | | AT5G45900 | |
| Gene (*Arabidopsis thaliana*) | SP1 | | AT1G63900 | |
| Gene (*Arabidopsis thaliana*) | PUB4 | | AT2G23140 | |
| Gene (*Arabidopsis thaliana*) | TOC132 | | AT2G16640 | |
| Gene (*Arabidopsis thaliana*) | TIC40 | | AT5G16620 | |
| Strain, strain background (*Arabidopsis thaliana*) | Col-0 | | | |
| Strain, strain background (*Agrobacterium tumefaciens*) | GV3101 | | | |
| Genetic reagent (*Arabidopsis thaliana*) | atg7-2 | PMID:20136727 | AT5G45900 | GABI_655B06 |
| Genetic reagent (*Arabidopsis thaliana*) | nbr1-1 | PMID:23341779 | AT4G24690 | SALK_135513 |
| Genetic reagent (*Arabidopsis thaliana*) | nbr1-2 | PMID:23341779 | AT4G24690 | GABI_246 H08 |
| Genetic reagent (*Arabidopsis thaliana*) | toc132-2 | PMID:15273297 | AT2G16640 | SAIL_667_04 |
| Genetic reagent (*Arabidopsis thaliana*) | tic40-4 | PMID:15659100 | AT5G16620 | SAIL_192_C10 |
| Genetic reagent (*Arabidopsis thaliana*) | sp1-2 | PMID:23118188 | AT1G63900 | SALK_063571 |
| Genetic reagent (*Arabidopsis thaliana*) | pub4-2 | PMID:26494759 | AT2G23140 | SALK_054373 |
| Genetic reagent (*Arabidopsis thaliana*) | Pro35S:mCherry-NBR1 | PMID:21606687 | AT4G24690 | |
| Genetic reagent (*Arabidopsis thaliana*) | ProUBQ10:mCherry-NBR1 | PMID:21606687 | AT4G24690 | |
| Genetic reagent (*Arabidopsis thaliana*) | ProNBR1:NBR1-GFP | PMID:28223514, 32967551 | AT4G24690 | |
| Genetic reagent (*Arabidopsis thaliana*) | Pro35S:RECA-GFP | PMID:9197266, 25649438 | | |
| Genetic reagent (*Arabidopsis thaliana*) | ProUBQ10:YFP-NBR1 | This study | AT4G24690 | See Methods and Materials Section 1 |
| Genetic reagent (*Arabidopsis thaliana*) | ProUBQ10:YFP-NBR1mPB | This study | AT4G24690 | See Methods and Materials Section 1 |
| Genetic reagent (*Arabidopsis thaliana*) | ProUBQ10:YFP-mAIM | This study | AT4G24690 | See Methods and Materials Section 1 |
| Genetic reagent (*Arabidopsis thaliana*) | ProUBQ10:YFP-NBR1DUBA2 | This study | AT4G24690 | See Methods and Materials Section 1 |
| Antibody | anti-NBR1 (Rabbit polyclonal) | PMID:31494674 | | EM IL (1:10) WB (1:1000) |
| Antibody | anti-TIC40 (Rabbit polyclonal) | Agrisera | Cat#: AS10709 | WB (1:2000) |
| Antibody | anti-PsbA/D1 (Rabbit polyclonal) | Agrisera | Cat#: AS05084 | WB (1:10,000) |
| Antibody | anti-LHCIIb (Rabbit polyclonal) | Agrisera | Cat#: AS01004 | WB (1:5000) |
| Recombinant DNA reagent | ProUBQ10:YFP-NBR | This study | AT4G24690 | See Materials and methods Section 1 |
| Recombinant DNA reagent | ProUBQ10:YFP-NBR1mPB | This study | AT4G24690 | See Materials and methods Section 1 |
| Software, algorithm | CLC main work bench 7 | Qiagen | | Cloning |
| Software, algorithm | Zen Software | Carl Zeiss | | Microscopy |
| Software, algorithm | Image J (Fiji) | NIH | | Image Quantification |

## Plant materials and growth conditions

*Arabidopsis thaliana* seeds of *atg7-2* (GABI_655B06) (*Chung et al., 2010*), *nbr1-1* (SALK_135513) (*Zhou et al., 2013*), *nbr1-2* (GABI_246H08) (*Zhou et al., 2013*), *atg7-2 nbr1-2* (*Jung et al., 2020*), *toc132-2* (SAIL_667_04) (*Kubis et al., 2004*), *tic40-4* (SAIL_192_C10) (*Kovacheva et al., 2005*), *Pro35S:mCherry-NBR1* (*Svenning et al., 2011*), ProUBQ10:mCherry-NBR1 (*Jung et al., 2020*), ProNBR1:NBR1-GFP (*Hafrén et al., 2017*; *Thirumalaikumar et al., 2021*), *Pro35S:RECA-GFP* (*Köhler et al., 1997*; *Spitzer et al., 2015*) were previously characterized. The *sp1-2* (SALK_063571) (*Ling et al., 2012*) and *pub4-2* (SALK_054373) (*Woodson et al., 2015*) mutant lines were acquired from *Arabidopsis* Biological Resource Center (https://abrc.osu.edu/) and *sp1-2 pub4-2 ProNBR1:NBR1-GFP* was generated by crossing. Primers used for genotyping the lines above are listed in *Supplementary file 1n*.

To fuse YFP to NBR1 mutant variants, NBR1, NBR1mPB, NBR1mAIM, and NBR1ΔUBA2 were cloned into the Gateway entry vector pDONR221 by the BP Clonase II reactions (Thermo Fisher Scientific) using Gateway expression vectors previously described (*Hafrén et al., 2017*). The resulting entry clones were recombined with pUBN-DEST-YFP (*Grefen et al., 2010*) via the LR Clonase II reaction (Thermo Fisher Scientific) to generate the expression vectors with YFP. The sequences were confirmed by Sanger sequencing with YFP and NBR1 primers. The expression vectors were introduced into *Agrobacterium tumefaciens* strain GV3101. *Agrobacterium* transformants were used to transform *nbr1-2* or *atg7-2 nbr1-2* mutants by the floral dip method (*Clough and Bent, 1998*). T1 plants were selected on the media supplemented with 10 mg/L Basta.

Seeds were surface-sterilized in 10% (v/v) bleach and 1% (v/v) Triton X-100 solution for 30 min and washed in distilled water at least five times. Seeds were sown on solid media containing 0.5 x Murashige & Skoog salts (MS), 1% (w/v) sucrose and 0.6% Phytagel and stratified at 4°C for 2–5 days before germination. Plants were grown in growth chambers at 22 °C under 16 hr of light (40 μmol m$^{-2}$ s$^{-1}$) and 8 hr of dark cycle (LL). For high-light treatment (HL), 8-d-old seedlings were exposed to 2000 W LED lights (1500 μmol m$^{-2}$ s$^{-1}$) at 12 °C for 2 hr followed by recovery for the indicated time.

## Transient expression in *Arabidopsis* leaf protoplasts

Isolation and transformation of *Arabidopsis* leaf protoplasts were performed as previously described (*Yoo et al., 2007*) with some modifications. Briefly, rosette leaves from 3-week-old *Arabidopsis* wild type (Col-0) and *tic40-4* (*Kovacheva et al., 2005*) plants were used for protoplast isolation. Protoplasts were released in enzyme solution (20 mM MES pH 5.7, 1.5% [w/v] cellulase R10, 0.4% [w/v] macerozyme R10, 0.4 M mannitol and 20 mM KCl, 10 mM CaCl2, and 0.1% BSA) for 1 hr and collected by centrifugation at 100 g for 5 min. Protoplasts were washed twice with W5 buffer (2 mM MES [pH 5.7] containing 154 mM NaCl, 125 mM CaCl2 and 5 mM KCl). Then, 7 μg of the *pUBN-YFP-NBR1mPB* vector was used for each transformation with polyethylene glycol. After transformation, the protoplasts were incubated at 22 °C under dark for 2 hr. For HL treatment, the transformed protoplasts were exposed to 1500 μmol m$^{-2}$ s$^{-1}$ at 12 °C for 2 hr followed by recovery in the dark at 22 °C. Control protoplasts were kept under dark conditions at 22 °C until imaging.

For confocal imaging, protoplasts were loaded onto an 18 Well Flat m-Slide (Ibidi). Images were captured on a Zeiss LSM 780 confocal microscope with a ×63 water immersion objective. YFP was excited with a 488 nm laser and detected using a 493–527 nm band-pass filter; chlorophyll was excited with a 633 nm laser and detected using a 642–695 nm band-pass filter. Between 9 and 10 protoplasts were used for quantification of each condition and phenotype.

## Light microscopy and image analysis

Confocal images were obtained in a Zeiss LSM 710 with a ×40 objective (LD C-Apochromat NA = 1.2 water immersion, Carl Zeiss). GFP, YFP, and chlorophyll were excited using a 488 nm laser and emission was collected from 450 to 560 nm for GFP/YFP and from 650 to 710 nm for chlorophyll. mCherry was excited using a 561 nm laser and emission collected from 570 to 640 nm. Quantification of confocal images was done with FIJI (*Schindelin et al., 2012*). To verify the specificity of the fluorescence signals, the emission spectra resulting from 488 nm excitation were collected between 420 and 720 nm using the lambda scan mode.

Multiphoton images were collected using a Nikon ×40 water-immersion objective lens (1.25 NA, CFI Apochromat Lambda S 40XC WI) on an Ultima IV multiphoton microscope (Bruker FM). Chlorophyll was imaged using 890 nm multiphoton excitation from an Insight laser (Spectra Physics) and

fluorescence emission was filtered using a dichroic cube filter set (720 nm, 630/69 nm, Chroma Technologies). Using manual estimation of leaf size and volumetric scanning from the surface to 100 microns deep, regions of interests were chosen and imaged. A hybrid photomultiplier tube (HPM-40, Becker&Hickl GmbH) detector was deployed in photon counting mode using a fast electronic board (SPC-150, Becker&Hickl GmbH), and Prairie View (Bruker FM) software. In presence of GFP markers, a second channel was imaged using a bialkali detector with 535/50 filter (Chroma Technologies).

The fluorescence images were made to 2D, using a maximum intensity projection and then 2D segmentation methods were applied to identify single chloroplasts. Cellpose 2.0.5 (Nucleus-model) with GPU acceleration (NVIDIA GeForce RTX 2080 Ti) generated robust chloroplast masks, which were then processed using Python (v3.9.12, Python Software Foundation) to calculate single chloroplast intensity and other morphological traits (*Stringer et al., 2021*).

## Protein preparation for western blots

Whole 8-day-old *Arabidopsis* seedlings were frozen with liquid nitrogen and homogenized in protein extraction buffer (150 mM Tris-HCl (pH7.5), 150 mM NaCl, 10 mM $MgCl_2$, 10% [v/v] glycerol, 2% [w/v] polyvinylpyrrolidone, 3 mM dithiothreitol, 2 mM phenylmethylsulfonyl fluoride, 0.1% [v/v] Triton X-100, 1 x protease inhibitor cocktail), and centrifuged at 25,000 *g* for 10 min at 4 °C. The supernatants were mixed with 0.25 volumes of 5 x SDS-PAGE sample buffer containing 10% [v/v] 2-mercaptoethanol and boiled before subjecting them to SDS-PAGE gel followed by immunoblotting with the indicated antibodies.

## Protein preparation for mass spectrometry (MS) analysis

Seven-day-old wild type (Col-0), *nbr1-2*, *atg7-2*, and *nbr1-2 atg7-2 Arabidopsis* seedlings were either grown under LL or left to recover for 24 hr after HL exposure as explained above. Whole seedlings were frozen in liquid nitrogen and grinded; protein extraction buffer (50 mM HEPES pH7.5, 5 mM $Na_2$ EDTA, 2 mM DTT, 1 x protease inhibitor cocktail) was added to the samples. After mixing, samples were left on ice for 15 min, and transferred to a homogenizer for gentle homogenization. The homogenate was left on ice for 15 min and centrifuged at 14,000 *g* for 1 min at 4 °C; 150 µL of the supernatant was transferred to clean 1.5 mL plastic tubes, mixed well by vortexing with 600 µL methanol and 150 µL chloroform. Then, 450 µL milliQ water was added to the sample and mixed by vortexing, followed by centrifugation at 14,000 *g* for 2 min. The top aqueous layer was removed and the proteins in the interphase collected with a pipette and transferred to a clean plastic tube followed by addition of 400 µL methanol, vortexing, and centrifugation at 14,000 *g* for 3 min. Methanol was removed from the tube without disturbing the pellet, which was left to dry in a vacuum concentrator.

## Liquid chromatography-tandem MS (LC-MS/MS)

Protein pellets were resuspended in 100 µL of 8 M urea. Then, 100 µg protein of each sample was reduced for 1 h at room temperate with 10 mM dithiothreitol, followed by alkylation with 20 mM iodoacetamide (IAA) for 1 hr. The reaction was quenched with 20 mM dithiothreitol (DTT) and diluted with 900 µL 25 mM ammonium bicarbonate to reduce the urea concentration below 1 M, digested overnight at 37 °C in the presence of 1.5 µg of sequencing grade modified trypsin (Promega). The resulting peptides were vacuum-dried in a vacuum concentrator to approximately 200 µL, acidified with 10% trifluoroacetic acid (TFA) (pH <3), desalted and concentrated on a 100 µL Bond Elut OMIX C18 pipette tip (Agilent Technologies A57003100) according to the manufacturer's instructions. Peptides were eluted in 50 µL of 75% acetonitrile, and 0.1% acetic acid, vacuum-dried, and resuspended in 15 µL 5% acetonitrile and 0.1% formic acid.

Nanoscale liquid chromatography (LC) separation of tryptic peptides was performed on a Dionex Ultimate 3000 Rapid Separation LC system (Thermo Fisher). Peptides were loaded onto a 20 µL nanoViper sample loop (Thermo Fisher) and separated on a C18 analytical column (Acclaim PepMap RSLC C18 column, 2 µm particle size, 100 Å pore size, 75 µm×25 cm, Thermo Fisher) by the application of a linear 2 hr gradient from 4% to 45% acetonitrile in 0.1% formic acid, with a column flow rate set to 250 nL/min. Analysis of the eluted tryptic peptides was performed online using a Q Exactive Plus mass spectrometer (Thermo Scientific) possessing a Nanospray Flex Ion source (Thermo Fisher) fitted with a stainless steel nano bore emitter operated in positive electrospray ionization (ESI) mode at a capillary voltage of 1.9 kV. Data-dependent acquisition of full MS scans within a mass range of

380–1500 m/z at a resolution of 70,000 was performed, with the automatic gain control (AGC) target set to $3.0\times10^6$, and the maximum fill time set to 200ms. High-energy collision-induced dissociation (HCD) fragmentation of the top eight most intense peaks was performed with a normalized collision energy of 28, with an intensity threshold of $4.0\times10^4$ counts and an isolation window of 3.0 m/z, excluding precursors that had an unassigned,+1 or>+7, charge state. MS/MS scans were conducted at a resolution of 17,500, with an AGC target of $2\times10^5$ and a maximum fill time of 300ms.

The resulting MS/MS spectra were analyzed using Proteome Discoverer software (version 2.5, Thermo Fisher), which was set up to search the *A. thaliana* proteome database, as downloaded from http://www.tair.com/ (Araport11_pep_20220914). Peptides were assigned using SEQUEST HT (**Eng et al., 1994**), with search parameters set to assume the digestion enzyme trypsin with a maximum of 1 missed cleavage, a minimum peptide length of 6, precursor mass tolerances of 10 ppm, and fragment mass tolerances of 0.02 Da. Carbamidomethylation of cysteine was specified as a static modification, while oxidation of methionine and N-terminal acetylation were specified as dynamic modifications. The target false discovery rate (FDR) of 0.01 (strict) was used as validation for peptide-spectral matches (PSMs) and peptides. Proteins that contained similar peptides and that could not be differentiated based on the MS/MS analysis alone were grouped to satisfy the principles of parsimony. Label-free quantification was performed in Proteome Discoverer as previously described (**Silva et al., 2006**) with a minimum Quan value threshold of 0.0001 using unique peptides, and '3 Top N' peptides used for area calculation. All samples were injected in two technical duplicates, and the protein abundances reflected the average of two technical replicates if proteins were detected in two technical replicates or used directly if the proteins were only detected in one technical replicate. Protein abundances were normalized using the median values of 150 proteins considered the least variable among each sample. The mass spectrometry proteomics data have been deposited to the ProteomeXchange Consortium *via* the PRIDE partner repository (**Perez-Riverol et al., 2019**) with the dataset identifier PXD039183.

Using the Perseus platform (**Tyanova et al., 2016**), intensity values from mass spectrometry were $log_2$ imputed and missing values were replaced with random numbers from a Gaussian distribution with a width of 0.3 and a downshift of 1.8. Statistical significance was determined using *t*-tests. Only proteins with at least two peptide spectral matches (one is the unique peptide) were selected for further analysis.

## Electron microscopy and immunogold labeling

Wild type (Col-0), *atg7-2*, and *nbr1-2* seedlings were germinated in liquid media containing 0.5 x MS and 1% sucrose. Eight-day-old cotyledons either grown under LL or at 24 hr after HL exposure were cut into small pieces and frozen in a high-pressure freezer (Leica EM Ice). To analyze the ultrastructure of chloroplasts, the samples were freeze-substituted in 2% (w/v) osmium tetroxide in acetone on dry ice overnight; samples were adjusted to room temperature on a rocker. After several rinses with acetone, the samples were infiltrated with Epon resin (Electron Microscopy Sciences) with increasing the concentration of Epon 10% (v/v), 25%, 50%, 75% in acetone, followed by three exchanges with 100% Epon. The samples were embedded and polymerized at 60 °C for 24 hr. For immunogold labeling, the high-pressure-frozen samples were freeze-substituted in 0.2% glutaraldehyde with 0.2% uranyl acetate in acetone at –90 °C in an automated freeze-substitution device (Leica AFS). After 3 days, the temperature was raised at 5 °C/hr to –60 °C and the samples were rinsed with precooled acetone three times and infiltrated with 30, 60, and 100% HM20 (Electron Microscopy Sciences) in acetone and polymerized under UV light at –50 °C. Sections were blocked with 5% (w/v) solution of nonfat milk in PBS (phosphate buffered saline) containing 0.1% Tween-20 (blocking solution) for 20 min, incubated with anti-NBR1 antibodies in the blocking solution (1:10) for 1 hr, rinsed three times with PBS containing 0.5% Tween-20, and incubated with anti-rabbit secondary antibody conjugated to gold particles (Electron Microscopy Sciences) in the blocking solution for 1 hr. After three rinses with the PBS containing 0.5% Tween-20 and another rinse with water, the samples were imaged with a transmission electron microscope (Thermo Fisher Scientific Talos).

## Chloroplast isolation

Intact chloroplasts were isolated as previously described with some modifications (**Kley et al., 2010**; **Lung et al., 2015**). Four-week-old leaves were punched repeatedly with a 1 ml pipette tip in buffer (0.3 M sorbitol, 50 mM HEPES/KOH [pH 7.5], 5 mM ethylenediaminetetraacetic acid [EDTA], 5 mM

ethyleneglycoltetraacetic acid [EGTA], 1 mM $MgCl_2$, 10 mM $NaHCO_3$, and 0.5 mM dithiothreitol) and filtered through cheesecloth. The filtrate was carefully loaded onto a two-step Percoll gradient that was prepared by overlaying 40% Percoll buffer on top of 85% Percoll and centrifuged for 20 min at 2,000 $g$ in a swing out rotor, brakes set off. The upper layer of the 40% Percoll containing broken chloroplasts was discarded, and the intact chloroplasts at the interface of the Percoll layers was collected and washed ive times by adding buffer and centrifuged for 5 min at 1000 $g$. Isolated chloroplasts were resuspended in buffer. We then added 0.25 volumes of 5 x SDS-PAGE sample buffer containing 10% (v/v) 2-mercaptoethanol to the samples. Protein extracts were subjected to SDS-PAGE followed by immunoblotting with the indicated antibodies.

## Antibodies

Antibodies against GFP (Chromotek), anti-LHCIIb (Agrisera), anti-cFBPase (Agrisera AS04043), anti-TIC40 (Agrisera), and anti-NBR1 (*Jung et al., 2020*) were obtained from the indicated sources.

## Statistical analyses

T-tests were performed in Microsoft Excel. ANOVA tests followed by post-hoc Tukey were performed using the calculator at https://astatsa.com/OneWay_Anova_with_TukeyHSD/. Data was visualized using GraphPad Prism 9 and Excel. The Venn diagram shown in *Figure 6* were created using http://bioinformatics.psb.ugent.be/webtools/Venn/.

## Accession numbers

NBR1 (At4g24690), ATG7 (At5g45900), SP1 (At1g63900), PUB4 (At2g23140), TOC132 (At2g16640), TIC40 (AT5G16620).

## Materials availability

Newly generated transgenic lines are available upon request.

# Acknowledgements

We thank Daniel Hofius for sharing NBR1 constructs with us, Taijoon Chung for providing mutant and transgenic lines, Janice Pennington for her assistance with the electron microscopy analysis, and Dr Sarah Swanson for her support with confocal imaging. This work was supported by grants from the U.S. National Science Foundation IOS-1840687 to MSO and RDV, U.S. Department of Energy grant DE-SC0019013 to MSO and KWE, National Institute of Health 1S10 OD026769-01 to MSO, NIH-NIGMS grant R01-GM124452 to RDV, and Australian Research Council (FL200100057) to AHM.

# Additional information

### Funding

| Funder | Grant reference number | Author |
|---|---|---|
| National Science Foundation | IOS-1840687 | Marisa S Otegui<br>Richard David Vierstra |
| U.S. Department of Energy | DE-SC0019013 | Marisa S Otegui<br>Kevin W Eliceiri |
| National Institutes of Health | 1S10 OD026769-01 | Marisa S Otegui |
| National Institutes of Health | R01-GM124452 | Richard David Vierstra |
| Australian Research Council | FL200100057 | A Harvey Millar |

The funders had no role in study design, data collection and interpretation, or the decision to submit the work for publication.

## Author contributions

Han Nim Lee, Conceptualization, Data curation, Formal analysis, Investigation, Visualization, Methodology, Writing – original draft, Writing – review and editing; Jenu Varghese Chacko, Conceptualization, Data curation, Formal analysis, Investigation, Visualization, Methodology, Writing – review and editing; Ariadna Gonzalez Solís, Conceptualization, Investigation, Visualization, Methodology, Writing – review and editing; Kuo-En Chen, Santiago Signorelli, Formal analysis, Investigation, Visualization, Methodology, Writing – review and editing; Jessica AS Barros, Formal analysis, Methodology, Writing – review and editing; A Harvey Millar, Richard David Vierstra, Conceptualization, Formal analysis, Supervision, Funding acquisition, Investigation, Methodology, Writing – review and editing; Kevin W Eliceiri, Conceptualization, Formal analysis, Supervision, Funding acquisition, Visualization, Methodology, Writing – review and editing; Marisa S Otegui, Conceptualization, Formal analysis, Supervision, Funding acquisition, Investigation, Visualization, Methodology, Writing – original draft, Project administration, Writing – review and editing

## Author ORCIDs

Han Nim Lee http://orcid.org/0000-0002-0429-6297
Jenu Varghese Chacko http://orcid.org/0000-0002-6676-0358
Richard David Vierstra http://orcid.org/0000-0003-0210-3516
Kevin W Eliceiri http://orcid.org/0000-0001-8678-670X
Marisa S Otegui http://orcid.org/0000-0003-4699-6950

## Decision letter and Author response

Decision letter https://doi.org/10.7554/eLife.86030.sa1
Author response https://doi.org/10.7554/eLife.86030.sa2

# Additional files

## Supplementary files

• Supplementary file 1. Proteome analysis by liquid chromatography-tandem mass spectrometry (LC-MS/MS). (a) Proteins identified by at least 2 peptide spectral matches.
(b) Normalized protein abundances based on the average of two technical replicates or used directly if the proteins were only detected in one technical replicate.
(c) Protein abundances expressed as Log2 values.
(d) Relative changes of protein abundance between LL and HL conditions in WT plants. Analysis was performed using the Perseus platform 2.0.6.0 (Tyanova et al., 2016), intensity values from MS/MS were log2 imputed and missing values were replaced with random numbers from a Gaussian distribution with a width of 0.3 and a downshift of 1.8. Statistical significance was determined using t-tests. The protein localizations and functions were categorized based on the GO term listed below. GO:0006914 (Autophagy), GO:0000502 (Proteasome), GO:0009507 (Chloroplast), GO:0005739 (Mitochondria), GO:0005777 (Peroxisome), GO:0005840 (Ribosome), GO:0009941 (Chloroplast envelope), GO:0009570 (Chloroplast stroma) and GO:0009534 (Chloroplast thylakoid).
(e) Relative changes of protein abundance between LL and HL conditions in the *atg7* mutant. Analysis was performed using the Perseus platform 2.0.6.0 (Tyanova et al., 2016), intensity values from MS/MS were log2 imputed and missing values were replaced with random numbers from a Gaussian distribution with a width of 0.3 and a downshift of 1.8. Statistical significance was determined using t-tests. The protein localizations and functions were categorized based on the GO term listed below. GO:0006914 (Autophagy), GO:0000502 (Proteasome), GO:0009507 (Chloroplast), GO:0005739 (Mitochondria), GO:0005777 (Peroxisome), GO:0005840 (Ribosome), GO:0009941 (Chloroplast envelope), GO:0009570 (Chloroplast stroma) and GO:0009534 (Chloroplast thylakoid).
(f) Relative changes of protein abundance between LL and HL conditions in the *nbr1* mutant. Analysis was performed using the Perseus platform 2.0.6.0 (Tyanova et al., 2016), intensity values from MS/MS were log2 imputed and missing values were replaced with random numbers from a Gaussian distribution with a width of 0.3 and a downshift of 1.8. Statistical significance was determined using t-tests. The protein localizations and functions were categorized based on the GO term listed below. GO:0006914 (Autophagy), GO:0000502 (Proteasome), GO:0009507 (Chloroplast), GO:0005739 (Mitochondria), GO:0005777 (Peroxisome), GO:0005840 (Ribosome), GO:0009941 (Chloroplast envelope), GO:0009570 (Chloroplast stroma) and GO:0009534 (Chloroplast thylakoid).
(g) Relative changes of protein abundance between LL and HL conditions in the *nbr1 atg7* double mutant. Analysis was performed using the Perseus platform 2.0.6.0 (Tyanova et al., 2016), intensity

values from MS/MS were log2 imputed and missing values were replaced with random numbers from a Gaussian distribution with a width of 0.3 and a downshift of 1.8. Statistical significance was determined using t-tests. The protein localizations and functions were categorized based on the GO term listed below. GO:0006914 (Autophagy), GO:0000502 (Proteasome), GO:0009507 (Chloroplast), GO:0005739 (Mitochondria), GO:0005777 (Peroxisome), GO:0005840 (Ribosome), GO:0009941 (Chloroplast envelope), GO:0009570 (Chloroplast stroma) and GO:0009534 (Chloroplast thylakoid).
(h) Comparison of protein abundances between WT and the *atg7* mutant under HL conditions. Analysis was performed using the Perseus platform 2.0.6.0 (Tyanova et al., 2016), intensity values from MS/MS were log2 imputed and missing values were replaced with random numbers from a Gaussian distribution with a width of 0.3 and a downshift of 1.8. Statistical significance was determined using t-tests. The protein localizations and functions were categorized based on the GO term listed below. GO:0006914 (Autophagy), GO:0000502 (Proteasome), GO:0009507 (Chloroplast), GO:0005739 (Mitochondria), GO:0005777 (Peroxisome), GO:0005840 (Ribosome), GO:0009941 (Chloroplast envelope), GO:0009570 (Chloroplast stroma) and GO:0009534 (Chloroplast thylakoid).
(i) Comparison of protein abundances between WT and the *nbr1* mutant under HL conditions. Analysis was performed using the Perseus platform 2.0.6.0 (Tyanova et al., 2016), intensity values from MS/MS were log2 imputed and missing values were replaced with random numbers from a Gaussian distribution with a width of 0.3 and a downshift of 1.8. Statistical significance was determined using t-tests. The protein localizations and functions were categorized based on the GO term listed below. GO:0006914 (Autophagy), GO:0000502 (Proteasome), GO:0009507 (Chloroplast), GO:0005739 (Mitochondria), GO:0005777 (Peroxisome), GO:0005840 (Ribosome), GO:0009941 (Chloroplast envelope), GO:0009570 (Chloroplast stroma) and GO:0009534 (Chloroplast thylakoid).
(j) Comparison of protein abundances between WT and the *nbr1 atg7* double mutant under HL conditions. Analysis was performed using the Perseus platform 2.0.6.0 (Tyanova et al., 2016), intensity values from MS/MS were log2 imputed and missing values were replaced with random numbers from a Gaussian distribution with a width of 0.3 and a downshift of 1.8. Statistical significance was determined using t-tests. The protein localizations and functions were categorized based on the GO term listed below. GO:0006914 (Autophagy), GO:0000502 (Proteasome), GO:0009507 (Chloroplast), GO:0005739 (Mitochondria), GO:0005777 (Peroxisome), GO:0005840 (Ribosome), GO:0009941 (Chloroplast envelope), GO:0009570 (Chloroplast stroma) and GO:0009534 (Chloroplast thylakoid).
(k) Comparison of protein abundances between WT and the *atg7* mutant under LL conditions. Analysis was performed using the Perseus platform 2.0.6.0 (Tyanova et al., 2016), intensity values from MS/MS were log2 imputed and missing values were replaced with random numbers from a Gaussian distribution with a width of 0.3 and a downshift of 1.8. Statistical significance was determined using t-tests. The protein localizations and functions were categorized based on the GO term listed below. GO:0006914 (Autophagy), GO:0000502 (Proteasome), GO:0009507 (Chloroplast), GO:0005739 (Mitochondria), GO:0005777 (Peroxisome), GO:0005840 (Ribosome), GO:0009941 (Chloroplast envelope), GO:0009570 (Chloroplast stroma) and GO:0009534 (Chloroplast thylakoid).
(l) Comparison of protein abundances between WT and the *nbr1* mutant under LL conditions. Analysis was performed using the Perseus platform 2.0.6.0 (Tyanova et al., 2016), intensity values from MS/MS were log2 imputed and missing values were replaced with random numbers from a Gaussian distribution with a width of 0.3 and a downshift of 1.8. Statistical significance was determined using t-tests. The protein localizations and functions were categorized based on the GO term listed below. GO:0006914 (Autophagy), GO:0000502 (Proteasome), GO:0009507 (Chloroplast), GO:0005739 (Mitochondria), GO:0005777 (Peroxisome), GO:0005840 (Ribosome), GO:0009941 (Chloroplast envelope), GO:0009570 (Chloroplast stroma) and GO:0009534 (Chloroplast thylakoid).
(m) Comparison of protein abundances between WT and the *nbr1 atg7* double mutant under LL conditions. Analysis was performed using the Perseus platform 2.0.6.0 (Tyanova et al., 2016), intensity values from MS/MS were log2 imputed and missing values were replaced with random numbers from a Gaussian distribution with a width of 0.3 and a downshift of 1.8. Statistical significance was determined using t-tests. The protein localizations and functions were categorized based on the GO term listed below. GO:0006914 (Autophagy), GO:0000502 (Proteasome), GO:0009507 (Chloroplast), GO:0005739 (Mitochondria), GO:0005777 (Peroxisome), GO:0005840 (Ribosome), GO:0009941 (Chloroplast envelope), GO:0009570 (Chloroplast stroma) and GO:0009534 (Chloroplast thylakoid).
(n) Primers used for genotyping.

• MDAR checklist

• Source data 1. Supplementary Data: Data used for all graphs presented in this study.

## Data availability

The mass spectrometry proteomics data have been deposited to the ProteomeXchange Consortium via the PRIDE partner repository with the dataset identifier PXD039183. All other data generated or analyzed during this study are included in the manuscript and supporting file.

The following dataset was generated:

| Author(s) | Year | Dataset title | Dataset URL | Database and Identifier |
|---|---|---|---|---|
| Chen K, Vierstra RD | 2023 | The autophagy receptor NBR1 directs the clearance of photodamaged chloroplasts | http://www.ebi.ac.uk/pride/archive/projects/PXD039183 | PRIDE, PXD039183 |

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
