## [Editor Report]

This fundamental work substantially advances our understanding of the role of the ubiquitin-binding NBR1 autophagy receptor protein. The work proposed a new function of this protein in the degradation of damaged chloroplasts upon exposing Arabidopsis plants to intense light. The evidence supporting the conclusions is compelling, using a combination of genetics, proteomics and state-of-the-art microscopy. The work will be of broad interest to cell biologists and biochemists.

---

## [Decision Letter]

**Decision letter after peer review:**

Thank you for submitting your article "The Autophagy Receptor NBR1 Directs the Clearance of Photodamaged Chloroplasts" for consideration by *eLife*. Your article has been reviewed by 3 peer reviewers, and the evaluation has been overseen by a Reviewing Editor and Jürgen Kleine-Vehn as the Senior Editor. The following individual involved in the review of your submission has agreed to reveal their identity: Cejudo Francisco Javier (Reviewer #1).

Essential revisions:

The reviewers emphasized that data showing that NBR1 decorates the exterior and interior of photodamaged chloroplasts is an important finding and categorize the results as valuable for the community. Essential revisions are:

1) The proteomic part is highlighted as the weakest point of the study. This part should be corrected – perhaps moved to supplementary as suggested by one reviewer or rewritten.

2) potential problems with the increased amount of NBR1 in atg7 mutant should be clarified and/or discussed.

3) the genetic data rely on single mutant alleles. This is not strictly rigorous in terms of interpreting the phenotypes thus conclusions should be tempered to reflect this.

*Reviewer #2 (Recommendations for the authors):*

Below is a list of a few rather mild comments.

1) line 63- SQTM1 change into SQSTM1.

2) lines 66-68 – I am not sure if the provided references contain information that the ZZ domain binds K48- or K63-linked polyubiquitinated proteins.

3) In some figures (e.g. Figure 2, Figure 3, Figure 7, Figure 8) there is no information about the number (N) of repeats used for plotting the charts and calculating of statistical significance of the differences. It would be useful to know how many chloroplasts were counted, from how many plants and/or how many cells.

4) line 602 should be "photodamaged".

5) line 654-655 (or other places in the discussion) – it would be worth including in the discussion the information (clearly shown in e.g. Jung et al. 2020, JXB) that the amount of NBR1 in atg7 mutant is increased in comparison to the wild type because of its defective degradation. The discussion of the effects of the single and double mutations (nbr1, atg7, and nbr1/atg7) should consider the above information. This could change the perspective because, in the atg7 mutant, the authors observed, in fact, the effect of the excess of NBR1 while in the double mutant, both proteins are missing.

*Reviewer #3 (Recommendations for the authors):*

In Figure 2, part G, the differences described in the text regarding specific protein levels in this immunoblot is not clearly seen. It would be good to provide quantification if possible. Also, it is not indicated how often such immunoblots were replicated, and maybe quantification could help to give an idea of reproducibility.

Figure 3 shows some very interesting data regarding the distribution (separate) of ATG8 and NBR1. The visual data are very clear, but I personally found the graphical data a little bit hard to assimilate. would it be an idea to provide a simplified schematic e.g. Venn diagram to simply convey the overlap (and lack thereof) of the populations in addition to these figures? Looking at this data I have been struck by the thought that whilst it is clear that there is reasonable discrimination between these two proteins associating different populations of chloroplasts, this might be a function of time relative to the initiated highlight stress. This is something the authors might like to comment on/investigate.

The authors have used single alleles for their mutant analysis, and I appreciate the pragmatic issues regarding using multiple alleles and/or complementation given that it increases the number of samples to be worked on significantly, however, strictly speaking, one cannot assume that all phenotypes shown in Figures 1 to 6 are truly linked to the mutations in question without this additional information. I would urge the authors to at least rein in their conclusions pertaining to these mutants. Of course, the effects seen are very likely (in terms of probability) to be linked to the mutations, but without additional genetic information, one really strictly cannot say this for certain. Given that the authors perform complementation experiments with tagged NBR1 shown in Figure 7, it would have been at least nice to see some data for complemented nbr1 mutant in the experiments presented in Figure 1 to Figure 6.

---

## [Author Response]

Essential revisions:The reviewers emphasized that data showing that NBR1 decorates the exterior and interior of photodamaged chloroplasts is an important finding and categorize the results as valuable for the community. Essential revisions are:1) The proteomic part is highlighted as the weakest point of the study. This part should be corrected – perhaps moved to supplementary as suggested by one reviewer or rewritten.

We have re-written the result section describing the proteomic data to make it more concise and clear. We have also modified Figure 6 and its two supplemental figure (Fig6supplemental figures 1 and 2) to make them more concise.

2) Potential problems with the increased amount of NBR1 in atg7 mutant should be clarified and/or discussed.

The increased levels of NBR1 in the *agt7* mutant and how this could influence the effects seen in the mutants studied in this manuscript is now discussed in lines 670-673. See also answers to reviewer #2.

3) The genetic data rely on single mutant alleles. This is not strictly rigorous in terms of interpreting the phenotypes thus conclusions should be tempered to reflect this.

We acknowledge the limitation of our conclusions for those results based on single mutant alleles in lines 630-631. See also answers to reviewer #3.

Reviewer #2 (Recommendations for the authors):Below is a list of a few rather mild comments.1) line 63- SQTM1 change into SQSTM1.

Fixed.

2) lines 66-68 – I am not sure if the provided references contain information that the ZZ domain binds K48- or K63-linked polyubiquitinated proteins.

We have changed this sentence to reflect what is contained in the cited references.

3) In some figures (e.g. Figure 2, Figure 3, Figure 7, Figure 8) there is no information about the number (N) of repeats used for plotting the charts and calculating of statistical significance of the differences. It would be useful to know how many chloroplasts were counted, from how many plants and/or how many cells.

We have now included in the figure legend the number of regions, chloroplasts, and plants analyzed in each case. To better appreciate the number of samples analyzed in each case, the graphs show the single data points. We have also submitted excel files with the raw data displayed in all graphs included in Figure 1 to 9 and supplemental figures.

4) line 602 should be "photodamaged".

Fixed.

5) line 654-655 (or other places in the discussion) – it would be worth including in the discussion the information (clearly shown in e.g. Jung et al. 2020, JXB) that the amount of NBR1 in atg7 mutant is increased in comparison to the wild type because of its defective degradation. The discussion of the effects of the single and double mutations (nbr1, atg7, and nbr1/atg7) should consider the above information. This could change the perspective because, in the atg7 mutant, the authors observed, in fact, the effect of the excess of NBR1 while in the double mutant, both proteins are missing.

The increased levels of NBR1 in the *agt7* mutant and how this could influence the effects seen in the mutants studied in this manuscript is now discussed in lines 670-673.

Reviewer #3 (Recommendations for the authors):In Figure 2, part G, the differences described in the text regarding specific protein levels in this immunoblot is not clearly seen. It would be good to provide quantification if possible. Also, it is not indicated how often such immunoblots were replicated, and maybe quantification could help to give an idea of reproducibility.

Following the reviewer suggestions, we have included in Figure 2 the ratios resulting from the quantification of the TIC40, cFBPase, and NBR1 western blots to show that, although both cFBPase and NBR1 are cytosolic proteins, cFBPase is partially depleted from the chloroplast fraction whereas NBR1 is enriched in chloroplast fractions under high light conditions in wild type and under low and high light conditions in the *atg7* mutant. The figure legend has been changed to include the explanation of this information. The figure shows a representative set of western blots. The experiment was repeated twice.

Figure 3 shows some very interesting data regarding the distribution (separate) of ATG8 and NBR1. The visual data are very clear, but I personally found the graphical data a little bit hard to assimilate. would it be an idea to provide a simplified schematic e.g. Venn diagram to simply convey the overlap (and lack thereof) of the populations in addition to these figures? Looking at this data I have been struck by the thought that whilst it is clear that there is reasonable discrimination between these two proteins associating different populations of chloroplasts, this might be a function of time relative to the initiated highlight stress. This is something the authors might like to comment on/investigate.

To make the graph easier to understand we have now clarified that the boxes show the percentage of chloroplasts labeled only with GFP-ATG8a, only with mCherry-NBR1, or both. As both GFP-ATG8a and mCherry-NBR1 are expressed on the same plants, the timing of the high light treatment should be similar for the chloroplast recruitment of the two proteins.

The authors have used single alleles for their mutant analysis, and I appreciate the pragmatic issues regarding using multiple alleles and/or complementation given that it increases the number of samples to be worked on significantly, however, strictly speaking, one cannot assume that all phenotypes shown in Figures 1 to 6 are truly linked to the mutations in question without this additional information. I would urge the authors to at least rein in their conclusions pertaining to these mutants. Of course, the effects seen are very likely (in terms of probability) to be linked to the mutations, but without additional genetic information, one really strictly cannot say this for certain. Given that the authors perform complementation experiments with tagged NBR1 shown in Figure 7, it would have been at least nice to see some data for complemented nbr1 mutant in the experiments presented in Figure 1 to Figure 6.

We agree with the reviewer and have addressed this comment above.